# Private Distribution Learning with Public Data:
# The View from Sample Compression*

**Shai Ben-David**
University of Waterloo
Vector Institute
shai@uwaterloo.ca

**Alex Bie**
University of Waterloo
yabie@uwaterloo.ca

**Clément L. Canonne**
University of Sydney
clement.canonne@sydney.edu.au

**Gautam Kamath**
University of Waterloo
Vector Institute
g@csail.mit.edu

**Vikrant Singhal**
University of Waterloo
vikrant.singhal@uwaterloo.ca

## Abstract

We study the problem of private distribution learning with access to public data. In this setup, which we refer to as *public-private learning*, the learner is given public and private samples drawn from an unknown distribution $p$ belonging to a class $\mathcal{Q}$, with the goal of outputting an estimate of $p$ while adhering to privacy constraints (here, pure differential privacy) only with respect to the private samples.

We show that the public-private learnability of a class $\mathcal{Q}$ is connected to the existence of a sample compression scheme for $\mathcal{Q}$, as well as to an intermediate notion we refer to as *list learning*. Leveraging this connection: (1) approximately recovers previous results on Gaussians over $\mathbb{R}^d$; and (2) leads to new ones, including sample complexity upper bounds for arbitrary $k$-mixtures of Gaussians over $\mathbb{R}^d$, results for agnostic and distribution-shift resistant learners, as well as closure properties for public-private learnability under taking mixtures and products of distributions. Finally, via the connection to list learning, we show that for Gaussians in $\mathbb{R}^d$, at least $d$ public samples are necessary for private learnability, which is close to the known upper bound of $d + 1$ public samples.

## 1 Introduction

Statistical analysis of sensitive data, and specifically parameter and density estimation, is a workhorse of privacy-preserving machine learning. To provide meaningful and rigorous guarantees on algorithm for these tasks, the framework of *differential privacy* (DP) [DMNS06] has been widely adopted by both algorithm designers and machine learning practitioners [App17, Abo18, XZA+23], and is, by and large, one of the past decade's success stories in principled approaches to private machine learning, with a host of results and implementations [Goo19a, Goo19b, HBAL19, Ope20] for many of the flagship private learning tasks.

Yet, however usable the resulting algorithms may be, DP often comes at a steep price: namely, many estimation tasks simple without privacy constraints provably require much more data to be performed privately; even more dire, they sometimes become *impossible* with any finite number of data points, absent some additional strong assumptions.

The prototypical example in that regard is learning a single $d$-dimensional Gaussian distribution from samples. In this task, a learner receives i.i.d. samples from an unknown $d$-dimensional Gaussian $p$

---

*Authors are listed in alphabetical order. Full paper: https://arxiv.org/abs/2308.06239.

37th Conference on Neural Information Processing Systems (NeurIPS 2023).

and is tasked with finding an estimate $q$ close to $p$ in total variation (TV) distance. Without privacy constraints, it is folklore that this can be done with $O(d^2)$ samples; yet once privacy enters the picture, in the form of pure differential privacy, *no finite sample algorithm for this task can exist*, unless a bound on the mean vector and covariance matrix are known.

This "cost of privacy" is, unfortunately, inherent to many estimation tasks, as the positive results (algorithms) developed over the years have been complemented with matching negative results (lower bounds). In light of these strong impossibility results, it is natural to wonder if one could somehow circumvent this often steep privacy cost by leveraging other sources of *public* data to aid the private learning process.

Recent work in finetuning machine learning models has tried to address the question whether, in situations where a vast amount of public data is available, one can combine the public data with a relatively small amount of *private* data to somehow achieve privacy guarantees for the private data and learning guarantees that would otherwise be ruled out by the aforementioned impossibility results. We, on the other hand, address the same question, but in the opposite setting, i.e., when the amount of public data available is much smaller than the amount of private data available. In other words, we answer the following question from new perspectives.

> *Can one leverage small quantities of public data to privately learn from sensitive data, even when private learning is impossible?*

This question was the focus of a recent study of Bie, Kamath, and Singhal [BKS22], the starting point of our work. In this paper, we make significant strides in this direction, by obtaining new "plug and play" results and connections in this public-private learning setting, and using them to obtain new sample complexity bounds for a range of prototypical density estimation tasks.

## 1.1 Our results

First, we establish a connection between learning (in the sense of *distribution learning* or *density estimation*) with public and private data (Definition 2.4) and *sample compression schemes for distributions* (Definition 2.5; see [ABDH$^+$20, Definition 4.2]), as well as an intermediate notion we refer to as *list learning* (Definition 3.2).

**Theorem 1.1** (Sample compression schemes, public-private learning, and list learning (Informal; see Theorem D.1)). *Let $\mathcal{Q}$ be a class of probability distributions and $m(\alpha, \beta)$ be a sample complexity function in terms of target error $\alpha$ and failure probability $\beta$. Then the following are equivalent.*

1. *$\mathcal{Q}$ has a sample compression scheme using $O(m(\alpha, \beta))$ samples.*

2. *$\mathcal{Q}$ is public-privately learnable with $O(m(\alpha, \beta))$ public samples.*

3. *$\mathcal{Q}$ is list learnable with $O(m(\alpha, \beta))$ samples.*

Despite its technical simplicity, this sample complexity equivalence turns out to be quite useful, and allows us to derive new public-private learners for an array of key distribution classes by leveraging known results on sample compression schemes. In particular, from the connection to sample compression schemes we are able to obtain new public-private learners for: (1) high-dimensional Gaussian distributions (Corollary E.1); (2) arbitrary mixtures of high-dimensional Gaussians (Theorem 1.2/Corollary E.2); (3) mixtures of public-privately learnable distribution classes (Theorem 4.3/E.5); and (4) products of public-privately learnable distribution classes (Theorem 4.3/E.7). For instance, the following is a consequence of the above connection.

**Theorem 1.2** (Public-private learning for mixtures of Gaussians (Informal; see Corollary E.2)). *The class of mixtures of $k$ arbitrary $d$-dimensional Gaussians is public-privately learnable with $m$ public samples and $n$ private samples, where*

$$m = \tilde{O}\left(\frac{kd}{\alpha}\right) \quad and \quad n = \tilde{O}\left(\frac{kd^2}{\alpha^2} + \frac{kd^2}{\alpha\varepsilon}\right),$$

*in which $\alpha$ is the target error and $\varepsilon$ the privacy parameter.*

We also examine public-private distribution learning in a setting with relaxed distributional assumptions, in which the distributions underlying the public and private data: (1) may differ (the case of

public-private *distribution shift*); and (2) may not be members of the reference class of distributions, and so we instead ask for error close to the best approximation of the private data distribution by a member of the class (the *agnostic* case). We show that *robust* sample compression schemes for a class of distributions can be converted into public-private learners in this *agnostic and distribution-shifted* setting. As a consequence, we have the following result for learning distributions that can be approximated by Gaussians under public-private distribution shift.

**Theorem 1.3** (Agnostic and distribution-shifted public-private learning for Gaussians (Informal; see Corollary F.1)). *There is a public-private learner that takes $m$ public samples and $n$ private samples from any pair of distributions $\tilde{p}$ and $p$ over $\mathbb{R}^d$ respectively, with $\mathrm{TV}(\tilde{p}, p) \leq \frac{1}{3}$, where*

$$m = O\left(d\right) \quad and \quad n = \tilde{O}\left(\frac{d^2}{\alpha} + \frac{d^2}{\alpha\varepsilon}\right),$$

*in which $\alpha$ is the target error and $\varepsilon$ the privacy parameter. With probability $\geq \frac{9}{10}$, the learner outputs $q$ with $\mathrm{TV}(q, p) \leq 3\mathrm{OPT} + \alpha$, where $\mathrm{OPT}$ is the total variation distance between $p$ and the closest $d$-dimensional Gaussian to it.*

Next, using the aforementioned connection to list learning, we are able to establish a fine-grained lower bound on the number of public data points required to privately learn high-dimensional Gaussians.

**Theorem 1.4** (Almost tight lower bound on privately learning Gaussians with public data (Informal; see Theorem 6.1)). *The class of all $d$-dimensional Gaussians is not public-privately learnable with fewer than $d$ public samples, regardless of the number of private samples.*

[BKS22] showed that $d$-dimensional Gaussians are public-privately learnable with $\tilde{O}(\frac{d^2}{\alpha^2} + \frac{d^2}{\alpha\varepsilon})$ private samples, *as soon as $d + 1$ public samples are available*. Thus, our result shows a very sharp threshold for the number of public data points necessary and sufficient to make private learning possible.

We also provide a general result for public-privately learning classes of distributions whose *Yatracos class* has finite VC dimension.

**Theorem 1.5** (VC dimension bound for public-private learning (Informal; see Theorem 7.2)). *Let $\mathcal{Q}$ be a class of probability distributions over a domain $\mathcal{X}$ such that the Yatracos class of $\mathcal{Q}$, defined as*

$$\mathcal{H} := \{\{x \colon f(x) > g(x)\} : f, g \in \mathcal{Q}\} \subseteq 2^{\mathcal{X}}$$

*has bounded VC dimension. Denote by $\mathrm{VC}(\mathcal{H})$ and $\mathrm{VC}^*(\mathcal{H})$ the VC and dual VC dimensions of $\mathcal{H}$ respectively. Then $\mathcal{Q}$ is public-privately learnable with $m$ public samples and $n$ private samples, where*

$$m = \tilde{O}\left(\frac{\mathrm{VC}(\mathcal{H})}{\alpha}\right) \quad and \quad n = \tilde{O}\left(\frac{\mathrm{VC}(\mathcal{H})^2 \mathrm{VC}^*(\mathcal{H})}{\varepsilon\alpha^3}\right),$$

*in which $\alpha$ is the target error and $\varepsilon$ the privacy parameter.*

The $\tilde{O}(\frac{\mathrm{VC}(\mathcal{H})}{\alpha})$ public sample requirement is less than the known $O(\frac{\mathrm{VC}(\mathcal{H})}{\alpha^2})$ sample requirement to learn with only public data via the non-private analogue of this result [Yat85, DL01].

## 1.2 Related work

The most closely related work is that of Bie, Kamath, and Singhal [BKS22], which initiated the study of distribution learning with access to both public and private data. That work studied algorithms for specific canonical distribution classes, while the present paper aims to broaden our understanding of public-private distribution learning in general, via connections to other problems and providing more general approaches for devising sample efficient public-private learners. In addition, we prove the first lower bounds on the amount of public data needed for private distribution learning.

There is a long line of work on private distribution learning, especially with regards to Gaussians and mixtures of Gaussians. [KV18] studied univariate Gaussians, showing that logarithmic dependencies on parameter bounds are necessary and sufficient in the case of pure DP, but can be removed under approximate DP. The same is true in the multivariate setting [KLSU19, KMS⁺22, TCK⁺22, AL22,

KMV22, LKO22]. For mixtures of Gaussians, most studies have focused on *parameter estimation* of mixture components [NRS07, KSSU19, CKM$^+$21, CCd$^+$23, AAL23b], and employ component separation and mixing weight assumptions. For *density estimation* (the setting studied in this work) [BKSW19, AAL21] give learnability results under various structural assumptions. The concurrent work of Azfali, Ashtiani, and Liaw [AAL23a] gives the first learnability result for general high-dimensional mixtures of Gaussians under approximate differential privacy.

Outside of distribution learning, there is a significant interest in using public data to improve private algorithms. Some problems studied include private query release, synthetic data generation, classification, mean estimation, empirical risk minimization, and stochastic convex optimization [JE13, BNS16, ABM19, NB20, BCM$^+$20, BMN20, LVS$^+$21, KADV23, LLHR23]. The definition of public-private algorithms that we adopt is from [BNS16], which studied classification in the PAC model. The VC dimension bound we give for public-private distribution learning relies on results from public-private classification [ABM19] and uniform convergence [BCM$^+$20].

On the technical side, we establish connections with distribution compression schemes as introduced by [ABDH$^+$20], and directly apply their results to establish new results for public-public learning. Related compression schemes for PAC learning for binary classification have been shown to be necessary and sufficient for learnability in those settings [LW86, MY16]. For further discussion of related work, please see Appendix A.

## 1.3 Limitations

Our work investigates the sample complexity of public-private learning, and does not give computationally efficient learners, or in some cases, *algorithmic* learners that run in finite time.[1] In particular, all public-private learners we obtain from sample compression run in time exponential in the sample complexity. Also, for our VC dimension upper bounds, we enumerate all realizable labellings of the input sample as per the relevant Yatracos class $\mathcal{H}$, which is not computable for general $\mathcal{H}$ [AABD$^+$20]. Finally, dependence on $\mathrm{VC}^*(\mathcal{H})$ in sample complexity is not ideal, as $\mathrm{VC}^*(\mathcal{H}) \leq 2^{\mathrm{VC}(\mathcal{H})+1} - 1$ is the best possible upper bound in terms of $\mathrm{VC}(\mathcal{H})$ for general $\mathcal{H}$ [Ass83].

## 2 Preliminaries

### 2.1 Notation

We denote by $\mathcal{X}$ the *domain of examples*. For a domain $\mathcal{U}$, denote by $\Delta(\mathcal{U})$ the set of all probability distributions over $\mathcal{U}$. We refer to a set $\mathcal{Q} \subseteq \Delta(\mathcal{X})$ as a *class of distributions over $\mathcal{X}$*.

We equip $\Delta(\mathcal{X})$ with the *total variation* metric, which is defined as follows: for $p, q \in \Delta(\mathcal{X})$, $\mathrm{TV}(p, q) := \sup_{B \in \mathcal{B}} |p(B) - q(B)|$, where $\mathcal{B}$ are the measurable sets of $\mathcal{X}$. For $p \in \Delta(\mathcal{X})$ and a set of distributions $L \subseteq \Delta(\mathcal{X})$, we denote their *point-set distance* by $\mathrm{dist}(p, L) := \inf_{q \in L} \mathrm{TV}(p, q)$.

We will let $\tilde{\boldsymbol{x}} = (\tilde{x}_1, \ldots, \tilde{x}_m) \in \mathcal{X}^m$ denote a *public dataset* and $\boldsymbol{x} = (x_1, \ldots, x_n) \in \mathcal{X}^n$ denote a *private dataset*. Their respective capital versions $\tilde{\boldsymbol{X}}$, $\boldsymbol{X}$ denote random variables for datasets realized by sampling from some underlying distribution. For $p \in \Delta(\mathcal{X})$, we denote by $p^m$ the distribution over $\mathcal{X}^m$ obtained by concatenating $m$ i.i.d. samples from $p$.

### 2.2 Public-private learning

**Definition 2.1** (Differential privacy [DMNS06]). Fix an input space $\mathcal{X}$ and an output space $\mathcal{Y}$. Let $\varepsilon, \delta > 0$. A randomized algorithm $\mathcal{A} : \mathcal{X}^n \to \Delta(\mathcal{Y})$ is $(\varepsilon, \delta)$-*differentially private ($(\varepsilon, \delta)$-DP)*, if for any private datasets $\boldsymbol{x}, \boldsymbol{x}' \in \mathcal{X}^n$ differing in one entry,

$$\mathbb{P}_{Y \sim \mathcal{A}(\boldsymbol{x})} \{Y \in B\} \leq \exp(\varepsilon) \cdot \mathbb{P}_{Y' \sim \mathcal{A}(\boldsymbol{x}')} \{Y' \in B\} + \delta \qquad \text{for all measurable } B \subseteq \mathcal{Y}.$$

---

[1]The comment of reviewer rJcv points out an approach to address the non-constructive nature of the reduction of public-privately learning mixture classes to public-privately learning the base class. To summarize: fixing the $m$ public samples and running the public-private learner on $n$ "null samples" repeatedly (with different random coins) produces a cover containing the true distribution with high probability. From here, we can run this procedure on all subsamples of the public data and construct a cover containing the true (mixture) distribution.

The case where $\delta = 0$ is referred to as *pure* differential privacy or $\varepsilon$-DP.

Our focus is on understanding the public data requirements for privately learning different classes of distributions. We seek to answer the following question:

> *For a class of distributions $\mathcal{Q}$, how much public data is necessary and sufficient to render $\mathcal{Q}$ privately learnable?*

To do so, we give the formal notion of "public-private algorithms" – algorithms that take public data samples and private data samples as input, and guarantee differential privacy with respect to the private data – as studied previously in the setting of binary classification [BNS16, ABM19]. We restrict our attention to public-private algorithms that offer a pure DP guarantee to private data.

**Definition 2.2** (Public-private $\varepsilon$-DP). Fix an input space $\mathcal{X}$ and an output space $\mathcal{Y}$. Let $\varepsilon > 0$. A randomized algorithm $\mathcal{A} \colon \mathcal{X}^m \times \mathcal{X}^n \to \Delta(\mathcal{Y})$ is *public-private $\varepsilon$-DP* if for any public dataset $\tilde{x} \in \mathcal{X}^m$, the randomized algorithm $\mathcal{A}(\tilde{x}, \cdot) : \mathcal{X}^n \to \Delta(\mathcal{Y})$ is $\varepsilon$-DP.

**Definition 2.3** (Public-private learner). Let $\mathcal{Q} \subseteq \Delta(\mathcal{X})$. For $\alpha, \beta \in (0, 1]$ and $\varepsilon > 0$, an $(\alpha, \beta, \varepsilon)$-*public-private learner for $\mathcal{Q}$* is a public-private $\varepsilon$-DP algorithm $\mathcal{A} : \mathcal{X}^m \times \mathcal{X}^n \to \Delta(\Delta(\mathcal{X}))$, such that for any $p \in \mathcal{Q}$, if we draw datasets $\tilde{X} = (\tilde{X}_1, ..., \tilde{X}_m)$ and $X = (X_1, ..., X_n)$ i.i.d. from $p$ and then $Q \sim \mathcal{A}(\tilde{X}, X)$,

$$\mathop{\mathbb{P}}_{\substack{\tilde{X} \sim p^m, X \sim p^n \\ Q \sim \mathcal{A}(\tilde{X}, X)}} \{\mathrm{TV}(Q, p) \leq \alpha\} \geq 1 - \beta.$$

**Definition 2.4** (Public-privately learnable class). We say that a class of distributions $\mathcal{Q} \subseteq \Delta(\mathcal{X})$ is *public-privately learnable with $m(\alpha, \beta, \varepsilon)$ public and $n(\alpha, \beta, \varepsilon)$ private samples* if for any $\alpha, \beta \in (0, 1]$ and $\varepsilon > 0$, there exists an $(\alpha, \beta, \varepsilon)$-public-private learner for $\mathcal{Q}$ that takes $m = m(\alpha, \beta, \varepsilon)$ public samples and $n = n(\alpha, \beta, \varepsilon)$ private samples.

When $\mathcal{Q}$ satisfies the above, we may omit the private sample requirement, and say that $\mathcal{Q}$ is *public-privately learnable with $m(\alpha, \beta, \varepsilon)$ public samples*.

Denote by $\mathrm{SC}_{\mathcal{Q}}(\alpha, \beta)$ the sample complexity of learning $\mathcal{Q}$. Our primary interest lies in determining when non-privately learnable $\mathcal{Q}$ can be public-privately learned with $m(\alpha, \beta, \varepsilon) = o(\mathrm{SC}_{\mathcal{Q}}(\alpha, \beta))$ public samples (at a target $\varepsilon$).

## 2.3 Sample compression schemes

One of the main techniques that we use in this work to create public-private learners for various distribution families is the *robust sample compression scheme* from [ABDH+20]. Roughly speaking, if every member $q$ of a class of distributions $\mathcal{Q}$ admits a way to encode enough information about itself in a small number of samples from $q$ and extra bits, such that it can be approximately reconstructed by a fixed and deterministic decoder, $\mathcal{Q}$ can be learned.

**Definition 2.5** (Robust sample compression [ABDH+20, Definition 4.2]). Let $r \geq 0$. We say $\mathcal{Q} \subseteq \Delta(\mathcal{X})$ admits $(\tau(\alpha, \beta), t(\alpha, \beta), m(\alpha, \beta))$ *r-robust sample compression* if for any $\alpha, \beta \in (0, 1]$, letting $\tau = \tau(\alpha, \beta)$, $t = t(\alpha, \beta)$, $m = m(\alpha, \beta)$, there exists a decoder $g \colon \mathcal{X}^\tau \times \{0, 1\}^t \to \Delta(\mathcal{X})$, such that the following holds:

For any $q \in \mathcal{Q}$ there exists an encoder $f_q \colon \mathcal{X}^m \to \mathcal{X}^\tau \times \{0, 1\}^t$ satisfying for all $x \in \mathcal{X}^m$ that for all $i \in [\tau]$, there exists $j \in [m]$ with $f_q(x)_i = x_j$, such that for every $p \in \Delta(\mathcal{X})$ with $\mathrm{TV}(p, q) \leq r$, if we draw a dataset $X = (X_1, ..., X_m)$ i.i.d. from $p$, then

$$\mathop{\mathbb{P}}_{X \sim p^m} \{\mathrm{TV}(g(f_q(X)), q) \leq \alpha\} \geq 1 - \beta.$$

When $\mathcal{Q}$ satisfies the above, we may omit the compression size complexity function $\tau(\alpha, \beta)$ and bit complexity function $t(\alpha, \beta)$, and say that $\mathcal{Q}$ is *r-robustly compressible with $m(\alpha, \beta)$ samples*. When $r = 0$, we say that $\mathcal{Q}$ admits $(\tau(\alpha, \beta), t(\alpha, \beta), m(\alpha, \beta))$ *realizable compression* and is *realizably compressible with $m(\alpha, \beta)$ samples*.

Both robust and realizable compression schemes satisfy certain useful properties that we use to develop public-private distribution learners in different settings. For example, the existence of a realizable compression scheme is closed under taking mixtures or products of distributions.

# 3 The connection to sample compression schemes

In this section we prove Theorem 1.1/D.1, the sample complexity equivalence between *sample compression* (Definition 2.5), *public-private learning* (Definition 2.4), and an intermediate notion we refer to as *list learning* (Definition 3.2). We do so by giving sample-efficient reductions between the three notions (Propositions 3.1, 3.4, and 3.5). The propositions state the quantitative translations between: compression size $\tau$ and number of bits $t$, number of private samples $n$, and list size $\ell$.

## 3.1 Compression implies public-private learning

We start by establishing that the existence of a sample compression scheme for $\mathcal{Q}$ implies the existence of a public-private learner for $\mathcal{Q}$.

**Proposition 3.1** (Compression $\implies$ public-private learning). *Let $\mathcal{Q} \subseteq \Delta(\mathcal{X})$. Suppose $\mathcal{Q}$ admits $(\tau(\alpha, \beta), t(\alpha, \beta), m_C(\alpha, \beta))$ realizable sample compression. Then $\mathcal{Q}$ is public-privately learnable with $m(\alpha, \beta, \varepsilon) = m_C(\frac{\alpha}{6}, \frac{\beta}{2})$ public and $n(\alpha, \beta, \varepsilon) = O((\frac{1}{\alpha^2} + \frac{1}{\alpha\varepsilon}) \cdot (t(\frac{\alpha}{6}, \frac{\beta}{2}) + \tau(\frac{\alpha}{6}, \frac{\beta}{2}) \log(m_C(\frac{\alpha}{6}, \frac{\beta}{2})) + \log(\frac{1}{\beta})))$ private samples.*

*Proof sketch.* The full proof is given in Appendix D.1, and mirrors that of Theorem 4.5 from [ABDH+20] (albeit adapted to the public-private setting). Fix $\alpha, \beta \in (0, 1]$ and $\varepsilon > 0$. Let $\tau = \tau(\frac{\alpha}{6}, \frac{\beta}{2})$, $t = t(\frac{\alpha}{6}, \frac{\beta}{2})$, and $m = m_C(\frac{\alpha}{6}, \frac{\beta}{2})$. We draw a public sample $\tilde{X}$ of size $m$, and enumerate all combinations of a size $\tau$ subset of $\tilde{X}$ with a binary string of length $t$. Essentially, we "guess" the encoding of the unknown distribution $p$. Running the decoder on this set of possible encodings gives us a set of distributions; some of them are close to $p$. Then, with private samples only, we use the pure DP 3-agnostic learner for finite classes (Fact C.2) to pick one such out. $\square$

## 3.2 Public-private learning implies list learning

Next, we show that the existence of a public-private learner for a class of distributions implies the existence of a *list learner* for the class. A list learner for a class $\mathcal{Q}$ takes input samples from any $p \in \mathcal{Q}$ and outputs a finite list of distributions, one of which is close to $p$.

**Definition 3.2** (List learner). *Let $\mathcal{Q} \subseteq \Delta(\mathcal{X})$. For $\alpha, \beta \in (0, 1]$ and $\ell \in \mathbb{N}$, an $(\alpha, \beta, \ell)$-list learner for $\mathcal{Q}$ is an algorithm $\mathcal{L} \colon \mathcal{X}^m \to \{L \subseteq \Delta(\mathcal{X}) : |L| \leq \ell\}$, such that for any $p \in \mathcal{Q}$, if we draw a dataset $\boldsymbol{X} = (X_1, \ldots, X_m)$ i.i.d. from $p$, then*

$$\mathbb{P}_{\boldsymbol{X} \sim p^m} \{\mathrm{dist}(p, \mathcal{L}(\boldsymbol{X})) \leq \alpha\} \geq 1 - \beta.$$

**Definition 3.3** (List learnable class). *A class of distributions $\mathcal{Q} \subseteq \Delta(\mathcal{X})$ is list learnable to list size $\ell(\alpha, \beta)$ with $m(\alpha, \beta)$ samples if for every $\alpha, \beta \in (0, 1]$, letting $\ell = \ell(\alpha, \beta)$ and $m = m(\alpha, \beta)$, there is an $(\alpha, \beta, \ell)$-list-learner for $\mathcal{Q}$ that takes $m$ samples.*

If $\mathcal{Q}$ satisfies the above, irrespective of the list size complexity $\ell(\alpha, \beta)$, we say $\mathcal{Q}$ is *list learnable with $m(\alpha, \beta)$ samples.*

Now, we state our reduction of list learning to public-private learning. The key step of our proof is showing that, upon receiving samples $\tilde{\boldsymbol{x}}$, outputting a finite cover of the list of distributions that a public-private learner would succeed on *given public data $\tilde{\boldsymbol{x}}$* is a successful strategy for list learning.

**Proposition 3.4** (Public-private learning $\implies$ list learning). *Let $\mathcal{Q} \subseteq \Delta(\mathcal{X})$. Suppose $\mathcal{Q}$ is public-privately learnable with $m_P(\alpha, \beta, \varepsilon)$ public and $n(\alpha, \beta, \varepsilon)$ private samples. Then for all $\varepsilon > 0$, $\mathcal{Q}$ is list-learnable to list size $\ell(\alpha, \beta) = \frac{10}{9} \exp(\varepsilon \cdot n(\frac{\alpha}{2}, \frac{\beta}{10}, \varepsilon))$ with $m(\alpha, \beta) = m_P(\frac{\alpha}{2}, \frac{\beta}{10}, \varepsilon)$ samples.*

*Proof.* Let $\varepsilon > 0$ be arbitrary. Fix any $\alpha, \beta \in (0, 1]$. By assumption, $\mathcal{Q}$ admits a $(\frac{\alpha}{2}, \frac{\beta}{10}, \varepsilon)$-public-private learner $\mathcal{A}$, which uses $m := m_P(\frac{\alpha}{2}, \frac{\beta}{10}, \varepsilon)$ public and $n := n(\frac{\alpha}{2}, \frac{\beta}{10}, \varepsilon)$ private samples. We use $\mathcal{A}$ to construct a $(\alpha, \beta, \frac{10}{9} \exp(\varepsilon n))$-list learner that uses $m$ samples. Consider any $\tilde{\boldsymbol{x}} = (\tilde{x}_1, \ldots, \tilde{x}_m) \in \mathcal{X}^m$ and the class

$$\mathcal{Q}_{\tilde{\boldsymbol{x}}} = \left\{ q \in \mathcal{Q} : \mathbb{P}_{\substack{\boldsymbol{X} \sim q^n \\ Q \sim \mathcal{A}(\tilde{\boldsymbol{x}}, \boldsymbol{X})}} \{\mathrm{TV}(Q, q) \leq \tfrac{\alpha}{2}\} \geq \tfrac{9}{10} \right\}.$$

Note that by definition, $\mathcal{Q}_{\tilde{\boldsymbol{x}}}$ has a $(\frac{\alpha}{2}, \frac{1}{10})$-learner under $\varepsilon$-DP that takes $n$ samples. Hence, by Fact C.1 it follows that any $\alpha$-packing of $\mathcal{Q}_{\tilde{\boldsymbol{x}}}$ must have size $\leq \frac{10}{9} \exp(\varepsilon n) =: \ell$. Let $\widehat{Q}_{\tilde{\boldsymbol{x}}}$ be such a maximal $\alpha$-packing, hence it is also an $\alpha$-cover of $\mathcal{Q}_{\tilde{\boldsymbol{x}}}$ with $|\widehat{Q}_{\tilde{\boldsymbol{x}}}| \leq \ell$. We define our list learner's output, $\mathcal{L}(\tilde{\boldsymbol{x}}) = \widehat{Q}_{\tilde{\boldsymbol{x}}}$. It remains to show that for any $p \in \mathcal{Q}$, with probability $\geq 1 - \beta$ over the sampling of $\tilde{\boldsymbol{X}} \sim p^m$, $\mathrm{dist}(p, \mathcal{L}(\tilde{\boldsymbol{X}})) \leq \alpha$. Suppose otherwise, that is, there exists $p_0 \in \mathcal{Q}$, such that

$$\mathbb{P}_{\tilde{\boldsymbol{X}} \sim p_0^m} \left\{ \mathrm{dist}(p_0, \mathcal{L}(\tilde{\boldsymbol{X}})) > \alpha \right\} > \beta.$$

Since $\mathcal{L}(\tilde{\boldsymbol{X}})$ is a $\alpha$-cover of $\mathcal{Q}_{\tilde{\boldsymbol{X}}}$, we have that with probability $> \beta$ over the sampling of $\tilde{\boldsymbol{X}} \sim p_0^m$, $p_0 \notin \mathcal{Q}_{\tilde{\boldsymbol{X}}}$. This contradicts the success guarantee of $\mathcal{A}$:

$$\mathbb{P}_{\substack{\tilde{\boldsymbol{X}} \sim p_0^m, \boldsymbol{X} \sim p_0^n \\ Q \sim \mathcal{A}(\tilde{\boldsymbol{X}}, \boldsymbol{X})}} \left\{ \mathrm{TV}(Q, p_0) > \frac{\alpha}{2} \right\} \geq \mathbb{P} \left\{ \mathrm{TV}(Q, p_0) > \frac{\alpha}{2} \,\Big|\, p_0 \notin \mathcal{Q}_{\tilde{\boldsymbol{X}}} \right\} \cdot \mathbb{P} \left\{ p_0 \notin \mathcal{Q}_{\tilde{X}} \right\}$$

$$> \frac{1}{10} \cdot \beta = \frac{\beta}{10}.$$

The second inequality follows by the definition of $Q_{\tilde{\boldsymbol{X}}}$: conditioned on the event $p_0 \notin \mathcal{Q}_{\tilde{\boldsymbol{X}}}$, the probability, over the private samples $\boldsymbol{X} \sim p_0^n$ and the randomness of the algorithm $\mathcal{A}$, that the output $Q$ of our algorithm satisfies $\mathrm{TV}(Q, p_0) \leq \frac{\alpha}{2}$ is $< \frac{9}{10}$. $\square$

### 3.3 List learning implies compression

We state the final component of Theorem 1.1/D.1: the existence of a list learner for a class of distributions $\mathcal{Q}$ implies the existence of a sample compression scheme for $\mathcal{Q}$. This follows from the definitions: given samples $\tilde{\boldsymbol{x}}$, the encoder of the sample compression scheme runs a list learner $\mathcal{L}$ on $\tilde{\boldsymbol{x}}$. It passes along $\boldsymbol{x}$, and, with knowledge of the target distribution $q$, the index $i$ of the distribution in $\mathcal{L}(\tilde{\boldsymbol{x}})$ that is close to $q$. The decoder receives this information and outputs $\mathcal{L}(\tilde{\boldsymbol{x}})_i$. The proof can be found in Appendix D.2.

**Proposition 3.5** (List learning $\implies$ compression). *Let $\mathcal{Q} \subseteq \Delta(\mathcal{X})$. Suppose $\mathcal{Q}$ is list learnable to list size $\ell(\alpha, \beta)$ with $m_L(\alpha, \beta)$ samples. Then $\mathcal{Q}$ admits $(\tau(\alpha, \beta), t(\alpha, \beta), m(\alpha, \beta)) = (m_L(\alpha, \beta), \log_2(\ell(\alpha, \beta)), m_L(\alpha, \beta))$ realizable sample compression.*

## 4 Applications

Here, we state a few applications of the connections obtained via Theorem 1.1/D.1. First, we recover and extend results on the public-private learnability of high-dimensional Gaussians and mixtures of Gaussians, using known results on sample compression schemes. Second, we describe the closure properties of public-private learnability: if a class $\mathcal{Q}$ is public-privately learnable, the class of mixtures of $\mathcal{Q}$ and the class of products of $\mathcal{Q}$ are also public-privately learnable.

### 4.1 Public-private learnability of Gaussians and mixtures of Gaussians

There are known realizable sample compression schemes for the class of Gaussians in $\mathbb{R}^d$, as well as for the class of all $k$-mixtures of Gaussians in $\mathbb{R}^d$ [ABDH+20]. Hence, these classes are public-privately learnable.

**Fact 4.1** (Robust compression scheme for Gaussians [ABDH+20, Lemma 5.3]). *The class of Gaussians over $\mathbb{R}^d$ admits $(O(d), O(d^2 \log(\frac{d}{\alpha})), O(d \log(\frac{1}{\beta})))$ $\frac{2}{3}$-robust sample compression.*

**Fact 4.2** (Realizable compression scheme for mixtures of Gaussians [ABDH+20, Lemma 4.8 applied to Lemma 5.3]). *The class of $k$-mixtures of Gaussians over $\mathbb{R}^d$ admits $(O(kd), O(kd^2 \log(\frac{d}{\alpha}) + \log(\frac{k}{\alpha})), O(\frac{kd \log(k/\beta) \log(1/\beta)}{\alpha}))$ realizable sample compression.*

**Gaussians.** From Theorem 1.1/D.1 and Fact 4.1, we get a public-private learner for Gaussians over $\mathbb{R}^d$ using $O(d \log(\frac{1}{\beta}))$ public and $\tilde{O}(\frac{d^2 + \log(1/\beta)}{\alpha^2} + \frac{d^2 + \log(1/\beta)}{\alpha \varepsilon})$ private samples (Corollary E.1). This recovers the result of [BKS22] on Gaussians up to a factor of $O(\log(\frac{1}{\beta}))$ in public sample complexity, and improves the private sample complexity by a $\mathrm{polylog}(1/\beta)$ factor.

**Mixtures of Gaussians.** From Theorem 1.1/D.1 and Fact 4.2, we get a public-private learner for the class of $k$-mixtures of Gaussians in $\mathbb{R}^d$ using $\tilde{O}(\frac{kd\log^2(1/\beta)}{\alpha})$ public and $\tilde{O}(\frac{kd^2+\log(1/\beta)}{\alpha^2} + \frac{kd^2+\log(1/\beta)}{\alpha\varepsilon})$ private samples (Theorem 1.2/Corollary E.2).

In terms of $k$, $d$, and $\alpha$, the public sample complexity of this learner is less than the $\tilde{\Theta}(\frac{kd^2}{\alpha^2})$ necessary and sufficient sample complexity for the problem non-privately [ABDH+20]. This also implies that in the regime where $\varepsilon > \alpha$, having more public samples (but still $\tilde{o}(\frac{kd^2}{\alpha^2})$) cannot improve private sample complexity. Under pure DP, no finite sample size suffices; under approximate DP, $\tilde{O}(\frac{k^2d^4\log(1/\delta)}{\alpha^2\varepsilon})$ has been shown to suffice [AAL23b].

Algorithms for *parameter estimation*, like the mixture of Gaussians estimators from [BKS22] cannot be directly compared as they target a strictly stronger success criteria under stronger assumptions on the underlying distribution. [BKS22] gives an estimator for separated Gaussian mixtures that uses $\tilde{O}(\frac{d\log(k)}{w_*})$ public and $\tilde{O}(\frac{d^2}{w_*\alpha^2} + \frac{d^2}{w_*\alpha^2})$ private samples satisfying $\frac{\varepsilon^2}{2}$-zCDP, where $w_*$ is the minimum component mixing weight. In terms of the privacy guarantee and private sample complexity, our result is a strict improvement for density estimation, since $k = O(\frac{1}{w_*})$. In the setting where $w_* = \Omega(\frac{1}{k})$, [BKS22]'s algorithm uses $\frac{1}{\alpha}$-factor fewer public samples. Furthermore, their algorithm runs in time polynomial in the sample complexity.

## 4.2 Public-private learnability of mixture and product distributions

If a class is realizably compressible, the class of its $k$-mixtures and the class of its $k$-products are also realizably compressible. Being realizably compressible with $O(m(\alpha, \beta))$ samples is equivalent to being public-privately learnable with $O(m(\alpha, \beta))$ public samples. Hence, we have black-box reductions of public-private learnability of mixture/product classes to public-private learnability of their base classes.

**Theorem 4.3** (Public-private learning for mixture and product distributions (Informal; see Theorems E.5 and E.7))**.** *Let $k \geq 1$. If $\mathcal{Q} \subseteq \Delta(\mathcal{X})$ is public-privately learnable with $m(\alpha, \beta, \varepsilon)$ public samples, then for any $\varepsilon_0 > 0$:*

1. *The class of $k$-mixtures of $\mathcal{Q}$ is public-privately learnable with $O(\frac{k\log(k/\beta)}{\alpha} \cdot m(\alpha, \beta, \varepsilon_0))$ public samples.*

2. *The class of $k$-products of $\mathcal{Q}$ (over $\mathcal{X}^k$) is public-privately learnable with $O(\log(\frac{k}{\beta}) \cdot m(\alpha/k, \beta, \varepsilon_0))$ public samples.*

The full statements and proof can be found in Appendix E.2. Here, we use the non-constructive reduction of list learning to public-private learning, and hence this does not yield a finite time algorithm. Note also that the target privacy $\varepsilon$ does not appear in the public sample complexity (it only affects the amount of private samples required).

## 5 Agnostic and distribution-shifted public-private learning

The setting we have examined thus far makes the following assumptions on the data generation process: (1) *same distribution* – public and private data are sampled from the same underlying distribution; and (2) *realizability* – public and private data are sampled from members of the class $\mathcal{Q}$.

[BKS22] shows that for Gaussians over $\mathbb{R}^d$, the first condition can be relaxed: they give an algorithm for the case where the public and the private data are generated from different Gaussians with bounded TV distance. However, they do not remove the second assumption.

We show that for general classes of distributions that *robust* compression schemes yield public-private learners, which: (1) can handle public-private distribution shifts (i.e., the setting where the public data and the private data distributions can be different); and (2) are agnostic, i.e., they do not require samples to come from a member of $\mathcal{Q}$, and instead, promise error close to the best approximation of the private data distribution by a member of $\mathcal{Q}$. Since Gaussians admit a robust compression scheme (Fact 4.1), we obtain public-private Gaussian learners that work under relaxed forms of these assumptions on the data generating process (Theorem 1.3/F.1).

We first formally define the notion of *agnostic and distribution-shifted public-private learning*, and then prove the main result of this section.

**Definition 5.1** (Agnostic and distribution-shifted public-private learner). Let $\mathcal{Q} \subseteq \Delta(\mathcal{X})$. For $\alpha, \beta \in (0,1], \varepsilon > 0, \gamma \in [0,1]$, and $c \geq 1$ a *$\gamma$-shifted $c$-agnostic $(\alpha, \beta, \varepsilon)$-public-private learner for* $\mathcal{Q}$ is a public-private $\varepsilon$-DP algorithm $\mathcal{A} : \mathcal{X}^m \times \mathcal{X}^n \to \Delta(\Delta(\mathcal{X}))$, such that for any $\tilde{p}, p \in \Delta(\mathcal{X})$ with $\mathrm{TV}(\tilde{p}, p) \leq \gamma$, if we draw datasets $\tilde{\boldsymbol{X}} = (\tilde{X}_1, \ldots, \tilde{X}_m)$ i.i.d. from $\tilde{p}$ and $\boldsymbol{X} = (X_1, \ldots, X_n)$ i.i.d. from $p$, and then $Q \sim \mathcal{A}(\tilde{\boldsymbol{X}}, \boldsymbol{X})$,

$$\mathop{\mathbb{P}}_{\substack{\tilde{\boldsymbol{X}} \sim \tilde{p}^m, \boldsymbol{X} \sim p^n \\ Q \sim \mathcal{A}(\tilde{\boldsymbol{X}}, \boldsymbol{X})}} \{\mathrm{TV}(Q, p) \leq c \cdot \mathrm{dist}(p, \mathcal{Q}) + \alpha\} \geq 1 - \beta.$$

**Theorem 5.2** (Robust compression $\implies$ agnostic and distribution-shifted public-private learning). *Let $\mathcal{Q} \subseteq \Delta(\mathcal{X})$ and $r > 0$. If $\mathcal{Q}$ admits $(\tau(\alpha, \beta), t(\alpha, \beta), m_C(\alpha, \beta))$ $r$-robust compression, then for every $\alpha, \beta \in (0,1]$ and $\varepsilon > 0$, there exists a $\frac{r}{2}$-shifted $\frac{2}{r}$-agnostic $(\alpha, \beta, \varepsilon)$-public-private learner for $\mathcal{Q}$ that uses $m(\alpha, \beta, \varepsilon) = m_C(\frac{\alpha}{12}, \frac{\beta}{2})$ public samples and $n(\alpha, \beta, \varepsilon) = O((\frac{1}{\alpha^2} + \frac{1}{\alpha \varepsilon}) \cdot (t(\frac{\alpha}{12}, \frac{\beta}{2}) + \tau(\frac{\alpha}{12}, \frac{\beta}{2}) \log(m_C(\frac{\alpha}{12}, \frac{\beta}{2})) + \log(\frac{1}{\beta})))$ private samples.*

*Proof.* The proof again mirrors the proof of Theorem 4.5 in [ABDH+20]. The key observation (and difference from the proof in Appendix D.1) is the following: for the unknown distribution $p \in \Delta(\mathcal{X})$, consider $\mathrm{dist}(p, \mathcal{Q})$. If $\mathrm{dist}(p, \mathcal{Q}) \geq \frac{r}{2}$, the output $Q$ of any algorithm satisfies $\mathrm{TV}(p, Q) \leq 1 \leq \frac{2}{r} \cdot \mathrm{dist}(p, \mathcal{Q})$. Hence, we can assume $\mathrm{dist}(p, \mathcal{Q}) < \frac{r}{2}$, and let $q_* \in \mathcal{Q}$ with $\mathrm{TV}(p, q_*) < \min\{\frac{r}{2}, \mathrm{dist}(p, \mathcal{Q}) + \frac{\alpha}{12}\}$ as guaranteed by such.

By triangle inequality, $\mathrm{TV}(\tilde{p}, q_*) < r$. This implies that when we generate hypotheses $\widehat{\mathcal{Q}}$ to choose from using the $r$-robust sample compression with samples from $\tilde{p}$, with high probability there will be some $q \in \widehat{\mathcal{Q}}$ with $\mathrm{TV}(q, q_*) \leq \frac{\alpha}{12}$. We have

$$\mathrm{TV}(p, q) \leq \mathrm{TV}(p, q_*) + \mathrm{TV}(q_*, q) \leq \mathrm{dist}(p, \mathcal{Q}) + \frac{\alpha}{12} + \frac{\alpha}{12} = \frac{\alpha}{6}.$$

Applying the 3-agnostic $\varepsilon$-DP learner for finite classes from [AAAK21] (Fact C.2) with the above setting of $n$ gives us the result. $\square$

# 6 Lower bounds

We give a lower bound on the number of public samples required to public-privately learn Gaussians in $\mathbb{R}^d$. We know that Gaussians in $\mathbb{R}^d$ are public-privately learnable with $d + 1$ public samples from [BKS22]. We show that this is within 1 of the optimal: the class of Gaussians in $\mathbb{R}^d$ is *not public-privately learnable* with $d - 1$ public samples. The following is the formal statement of Theorem 1.4.

**Theorem 6.1.** *The class $\mathcal{Q}$ of all Gaussians in $\mathbb{R}^d$ is* not *public-private learnable with $m_P(\alpha, \beta, \varepsilon) = d - 1$ public samples, regardless of the number of private samples. That is, there exists $\alpha_d, \beta_d > 0$ such that for any $n \in \mathbb{N}$, $\mathcal{Q}$ does not admit a $(\alpha_d, \beta_d, 1)$-public-private learner using $d - 1$ public and $n$ private samples.*

Our result leverages the connection between public-private learning and list learning. The existence of such a public-private learner described above would imply the existence of a list learner for $d$-dimensional Gaussians taking $d - 1$ samples as input. We show, using a "no-free-lunch"-style argument (e.g. Theorem 5.1 from [SSBD14]) that such a list learner cannot exist. The proof of Theorem 6.1, given in Appendix G, goes through the following steps.

1. We reduce list learning to public-private learning, via Proposition 3.4;

2. We establish a technical lemma that relates the PAC guarantee of a list learner with its average performance over a set of problem instances, via a "no-free-lunch"-style argument (Lemma G.1);

3. For every $d \geq 2$, we find a sequence of hard subclasses of Gaussians over $\mathbb{R}^d$, which satisfy the conditions of Lemma G.1. This forms the set of hard problem instances that imply a lower bound on the error of any list learner for the class (does not receive enough samples);

4. Since list learning to arbitrary error with few samples is impossible, public-private learning to arbitrary error with few public samples must also be impossible.

# 7 Learning when the Yatracos class has finite VC dimension

In this section, we describe a public-private learner for classes of distributions whose *Yatracos class* has finite VC dimension. We start by defining the Yatracos class of a family of distributions.

**Definition 7.1** (Yatracos class)**.** For $\mathcal{Q} \subseteq \Delta(\mathcal{X})$, the *Yatracos class* of $\mathcal{Q}$ is given by

$$\mathcal{H} = \{\{x \in \mathcal{X} : p(x) > q(x)\} : p \neq q \in \mathcal{Q}\}.\text{[2]}$$

**Theorem 7.2.** *Let $\mathcal{Q} \subseteq \Delta(\mathcal{X})$. Let $\mathcal{H}$ be the Yatracos class of $\mathcal{Q}$, denote by $\mathrm{VC}(\mathcal{H})$ and $\mathrm{VC}^*(\mathcal{H})$ the* VC *and dual* VC *dimension of $\mathcal{H}$. $\mathcal{Q}$ is public-privately learnable with $m$ public and $n$ private samples, where*

$$m = O\left(\frac{\mathrm{VC}(\mathcal{H})\log\left(\frac{1}{\alpha}\right) + \log\left(\frac{1}{\beta}\right)}{\alpha}\right) \quad \text{and} \quad n = O\left(\frac{\mathrm{VC}(\mathcal{H})^2\,\mathrm{VC}^*(\mathcal{H}) + \log(\frac{1}{\beta})}{\varepsilon\alpha^3}\right).$$

Theorem 7.2/1.5 says that classes of distributions whose Yatracos class have finite VC dimension can be public-privately learned. Note that the number of public samples used is indeed fewer than the $O(\frac{\mathrm{VC}(\mathcal{H})}{\alpha^2})$ sample requirement in the non-private analogue of the result (Fact H.1).

The proof is given in Appendix H. The result is a consequence of a known public-private uniform convergence result [BCM$^+$20, Theorem 10]. To adapt it to our setting, we (1) modify their result for pure DP (rather than approximate DP); and (2) conclude that uniform convergence over the Yatracos sets of $\mathcal{Q}$ suffices to implement the learner from Fact H.1.

# 8 Conclusion

In this work, we connect public-private distribution learning to the notions of sample compression and list learning. In doing so, for broad classes of distributions, we introduce approaches to: (1) design sample-efficient public-private learners; (2) prove lower bounds on how much public data is required for public-private learnability. In the following, we list several questions for future study.

**Question 8.1.** *For a class $\mathcal{Q}$, our work examines the minimal amount of public data needed to render $\mathcal{Q}$ pure privately learnable. How much do we need to render $\mathcal{Q}$ approximate DP learnable?*

**Question 8.2.** *The VC bound of Theorem 1.5/7.2, although more general, is "qualitatively" loose. For Gaussians, it yields a $O(\frac{d^2}{\alpha})$ public sample complexity, though we know $O(d)$ is possible, notably with no dependence on $\alpha$. On the other hand, $\Omega(1/\alpha)$ public samples are required for public-privately learning mixtures of Gaussians. What qualities of a distribution admit public-private learning with $\alpha$-independent public sample complexity? Stronger: can we find more illuminating characterizations of the sample complexity of list learning?*

**Question 8.3.** *Our work studies sample complexity improvements from using public data. Can public data lead to algorithmic improvements, that is, runtime efficiency or simpler algorithms?*

## Acknowledgments and Disclosure of Funding

We thank the anonymous reviewers for their helpful feedback.

AB, GK, and VS acknowledge funding from an NSERC discovery grant. AB is supported by a David R. Cheriton Graduate Scholarship. CC is supported by an ARC DECRA and an unrestricted gift from Google. GK is supported by a Canada CIFAR AI Chair, an unrestricted gift from Google, a University of Waterloo startup grant, and an unrestricted gift from Apple.

---

[2]This is for when the distributions in $\mathcal{Q}$ are discrete. For classes of continuous distributions, we substitute $p$ and $q$ for their respective density functions.

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

## A  Extended related work

**Privately learning Gaussians.**  Our work studies the task of learning arbitrary, unbounded Gaussians while offering differential privacy guarantees.  Basic private algorithms for the task (variants of "clip-and-noise") impose boundedness assumptions on the underlying parameters of the unknown Gaussian, since their sample complexities grow to infinity as the bounds widen to include more allowed distributions.  Understanding these dependencies without public data has been a topic of significant study. [KV18] examined univariate Gaussians, showing that logarithmic dependencies on parameter bounds are necessary and sufficient in the case of pure DP, but can be removed under approximate DP. The same is true in the multivariate setting [AAAK21, KMS+22, TCK+22, AL22, KMV22, LKO22]. [BKS22] shows that instead of relaxing to approximate DP to handle arbitrary Gaussians, one can employ a small amount of public data; our lower bound tells us almost exactly how much is needed.  Furthermore, our reductions between public-private learning and list learning offers the conclusion that the role of public data is precisely for bounding: distribution classes that can be privately learned with a small amount of public data *are exactly* the distributions that can be bounded with a small amount of public data.

**Privately learning mixtures of Gaussians.**  Another line of related work is that on privately learning mixtures of Gaussian, but without any public data. [NRS07] provided a subsample-and-aggregate approach to learn the parameters of mixtures of spherical Gaussians based on the work by [VW02] under the weaker, approximate DP. Recently, [CCd+23] improved on this by weakening the separation condition required for the mixture components. [KSSU19] provided the first polynomial-time, approximate DP algorithms to learn the parameters of mixtures of non-spherical Gaussians under weak boundedness assumptions. [CKM+21] improved on their work both in terms of the sample complexity and the separation assumption. [AAL23b] provided a polynomial-time reduction for privately and efficiently learning mixtures of unbounded Gaussians from the approximate DP setting to its non-private counterpart (albeit at a polynomial overhead in the sample complexity).

Our work falls into the category of private density estimation, for which [AAL21] gave new algorithms for the special case of spherical Gaussians under approximate DP, while [BKSW19] gave (computationally inefficient) algorithms for bounded Gaussians under pure DP. For comparison, the latter would have infinite sample complexity for unbounded Gaussians, but our work provides finite private sample complexity even under pure DP using public data. On the other hand, [ASZ21] showed hardness results for privately learning mixtures of Gaussians with known covariances. The concurrent work of Azfali, Ashtiani, and Liaw [AAL23a] gives the first learnability result for general high-dimensional mixtures of Gaussians under approximate differential privacy.

**Theory for private algorithms with public data.**  Beyond distribution learning, there is a lot more work that investigates how public data can be employed in private data analysis. Some specific areas include private query release, synthetic data generation, and classification [JE13, BNS16, ABM19, NB20, BCM+20, BMN20, LVS+21, KADV23], and the results are a mix of theoretical versus empirical. The definition of public-private algorithms that we adopt is from [BNS16], which studied classification in the PAC model. The VC dimension bound we give for public-private distribution learning relies on results from public-private classification [ABM19] and uniform convergence [BCM+20].

A concurrent and independent work [LLHR23] also studies learning with public and private data, focusing on the problems of mean estimation, empirical risk minimization, and stochastic convex optimization. The focus of the two works is somewhat different, both in terms of the type of problems considered (we study density estimation) and the type of results targeted. That is, our objective is to draw connections between different learning concepts and exploring the resulting implications for public-private distribution learning, while their goal seems to be understanding the precise error rates for some fundamental settings.

**Private machine learning with public data.**  Within the context of private machine learning, there has been significant interest in how to best employ public data. The most popular method is pretraining [ACG+16, PCS+19, TB21, LWAFF21, YZCL21, LTLH22, YNB+22, GHN+23, HASP22] (though some caution about this practice [TKC22]), while other methods involve computing statistics about the private gradients [ZWB21, YZCL21, KRRT21, AGM+22, GKW23], or training a student

model [PAE$^+$17, PSM$^+$18, BTGT18]. For more discussion of public data for private learning, see Section 3.1 of [CDE$^+$23].

**List learning** We also use the notion of *list learning* in this work, which is a "non-robust" version of the well-known *list-decodable learning* [AAL21, RY20, BBV08, CSV17, DKS18, KS17], where the goal is still to output a list of distributions that contains one that is accurate with respect to the true distribution, but the sampling may happen from a corrupted version of the underlying distribution.

**Hybrid differential privacy.** Finally, a related setting is the hybrid model, in which samples require either local or central differential privacy [AKZ$^+$17]. Some learning tasks studied in this model include mean estimation [ADK20] and transfer learning [KS22].

# B Technical overview

**New connections for public-private learning.** Our first contribution is establishing connections between sample compression, public-private learning, and list learning, via reductions. We show that: (1) sample compression schemes yield public-private learners; (2) public-private learners yield list learners; and (3) list learners yield sample compression schemes.

(1) and (3) are straightforward to prove: (1) *compression implies public-private learning* follows from a modification of an analogous result of [ABDH$^+$20], where we observe that in their proof, the learner's two-stage process of drawing a small *compression sample* used to generate a finite set of hypotheses using the compression scheme, followed by *hypothesis selection* with a larger sample, can be cleanly divided into using public and private samples respectively. In the latter stage, we employ a known pure DP hypothesis selection algorithm [BKSW19, AAAK21]). (3) *List learning implies compression* follows immediately from the definitions.

For (2) *public-private learning implies list learning*, we show non-constructively that there exists a list learner for a class, given a public-private learner for that class, but do not provide an algorithmic translation of the public-private learner to a list learner. For a set of samples $S$, we show that outputting a finite cover of the list of distributions on which the public-private learner succeeds on, when using $S$ as the public samples, is a correct output for a list learner. Hence the list learner we construct, on input $S$, outputs this finite cover as determined by (but not explicitly constructed from) the public-private learner.

**Agnostic and distribution-shifted public-private learning.** We identify some distributional assumptions that can be relaxed in the public-private learning setup. We obtain agnostic and distribution-shifted public-private learners via a connection to robust compression schemes. The reduction ideas are similar to those in the case of public-private learning and non-robust sample compression described above.

**Lower bound on public-private learning of Gaussians.** Our lower bound for public-privately learning high-dimensional Gaussians exploits our above connection between public-private learning and list learning, and applies a "no-free-lunch"-style argument. The latter uses the fact that an algorithm's worst-case performance cannot be better than its performance when averaged across all the problem instances. In other words, we have two main steps in our proof: (1) we first claim that due to our above reduction, a lower bound for list learning high-dimensional Gaussians would imply a lower bound on the public sample complexity for public-privately learning high-dimensional Gaussians; and (2) assuming certain accuracy guarantees and a sample complexity for our list learner for high-dimensional Gaussians; we show that across a set of adversarially chosen problem instances, our average accuracy guarantee fails, which is a contradiction to our assumed worst-case guarantees.

We observe that the lower bound from Theorem 6.1 establishes that at least $d$ public samples are necessary for public-private learning to vanishingly small error as $d$ increases (impossibility of learning to a target error, via the application of Lemma G.1, is related to the bound on $\eta$ from Equation (6), which decreases exponentially with $d$). A natural question is whether the result can be strengthened to say that there is a single target error, simultaneously for all $d$, for which learning is impossible without $d$ public samples.

**VC dimension upper bound for public-private learning.** Our proof involves invoking the existing results for public-private binary classification [ABM19] and uniform convergence [BCM+20]. We use them to implement Yatracos' minimum distance estimator [Yat85, DL01] in a public-private way.

## C  Extended preliminaries

### C.1  Notation

**Covers and packings.** For $\alpha > 0$ and $\mathcal{Q} \subseteq \Delta(\mathcal{X})$, we say that $\mathcal{C} \subseteq \Delta(\mathcal{X})$ is an $\alpha$-*cover* of $\mathcal{Q}$ if for any $q \in \mathcal{Q}$, there exists a $p \in \mathcal{C}$ with $\mathrm{TV}(p, q) \leq \alpha$. For $\alpha > 0$ and $\mathcal{Q} \subseteq \Delta(\mathcal{X})$, we say that $\mathcal{P} \subseteq \mathcal{Q}$ is an $\alpha$-*packing* of $\mathcal{Q}$ if for any $p \neq q \in \mathcal{P}$, $\mathrm{TV}(p, q) > \alpha$.

**Class of $k$-mixtures.** Let $\mathcal{Q} \subseteq \Delta(\mathcal{X})$ be a class of distributions. For any $k \geq 1$, the *class of $k$-mixtures of $\mathcal{Q}$* is given by

$$\mathcal{Q}^{\oplus k} := \left\{ \sum_{i=1}^{k} w_i q_i : q_i \in \mathcal{Q}, w_i \geq 0 \text{ for all } i \in [k] \text{ and } \sum_{i=1}^{k} w_i = 1 \right\}.$$

**Class of $k$-products.** Let $\mathcal{Q} \subseteq \Delta(\mathcal{X})$ be a class of distributions over $\mathcal{X}$. For any $k \geq 1$, $q = (q_1, \ldots, q_k)$ is a product distribution over $\mathcal{X}^k$, if $q_i \in \mathcal{Q}$ for all $i \in [k]$ and for $X \sim q$, the $i$-th component $X_i$ of $X$ is independently (of all the other coordinates) sampled from $q_i$. The *class of $k$-products of $\mathcal{Q}$ over $\mathcal{X}^k$* is given by

$$\mathcal{Q}^{\otimes k} := \{ (q_1, \ldots, q_k) : q_i \in \mathcal{Q} \text{ for all } i \in [k] \}.$$

### C.2  Privacy

The following is a known hardness result on density estimation of distributions under pure differential privacy, and is based on the standard "packing lower bounds".

**Fact C.1** (Packing lower bound [BKSW19, Lemma 5.1]). *Let $\mathcal{Q} \subseteq \Delta(\mathcal{X})$, $\alpha \in (0, 1]$, and $\varepsilon > 0$. Let $\widehat{\mathcal{Q}}$ be any $\alpha$-packing of $\mathcal{Q}$. Any $\varepsilon$-DP algorithm $\mathcal{A} \colon \mathcal{X}^n \to \Delta(\Delta(\mathcal{X}))$ that, upon receiving $n$ i.i.d. samples $X_1, \ldots, X_n$ from any $p \in \mathcal{Q}$, outputs $Q$ with $\mathrm{TV}(Q, p) \leq \frac{\alpha}{2}$ with probability $\geq \frac{9}{10}$ requires*

$$n \geq \frac{\log(|\widehat{\mathcal{Q}}|) - \log(\frac{10}{9})}{\varepsilon}.$$

The next result guarantees the existence of agnostic learners for finite hypothesis classes under pure differential privacy.

**Fact C.2** (Pure DP 3-agnostic learner for finite $\mathcal{Q}$ [BKSW19], [AAAK21, Theorem 2.24]). *Let $\mathcal{Q} \subseteq \Delta(\mathcal{X})$ with $|\mathcal{Q}| < \infty$. For every $\alpha, \beta \in (0, 1]$ and $\varepsilon > 0$, there exists an $\varepsilon$-DP algorithm $\mathcal{A} \colon \mathcal{X}^n \to \Delta(\Delta(\mathcal{X}))$, such that for any $p \in \Delta(\mathcal{X})$, if we draw a dataset $\boldsymbol{X} = (X_1, \ldots, X_n)$ i.i.d. from $p$ and then $Q \sim \mathcal{A}(\boldsymbol{X})$,*

$$\mathbb{P}_{\substack{\boldsymbol{X} \sim p^n \\ Q \sim \mathcal{A}(\boldsymbol{X})}} \{ \mathrm{TV}(Q, p) \leq 3 \cdot \mathrm{dist}(p, \mathcal{Q}) + \alpha \} \geq 1 - \beta,$$

*where*

$$n = O\left( \frac{\log(|\mathcal{Q}|) + \log(\frac{1}{\beta})}{\alpha^2} + \frac{\log(|\mathcal{Q}|) + \log(\frac{1}{\beta})}{\alpha \varepsilon} \right).$$

## D  Statements and proofs for Section 3 – The connection to sample compression schemes

The following is the full formal statement of Theorem 1.1.

**Theorem D.1** (Sample complexity equivalence between sample compression, public-private learning, and list learning). *Let $\mathcal{Q} \subseteq \Delta(\mathcal{X})$. Let $m : (0, 1]^2 \to \mathbb{N}$ be a sample complexity function, such that $m(\alpha, \beta) = \mathrm{poly}(\frac{1}{\alpha}, \frac{1}{\beta})$.[3] Then the following are equivalent.*

---

[3]The reductions between the learners do not need this assumption, it is only used to state the sample complexity equivalence.

1. $\mathcal{Q}$ *is realizably compressible with* $m_C(\alpha, \beta) = O(m(\alpha, \beta))$ *samples.*

2. $\mathcal{Q}$ *is public-privately learnable with* $m_P(\alpha, \beta, \epsilon) = O(m(\alpha, \beta))$ *public samples.*

3. $\mathcal{Q}$ *is list learnable with* $m_L(\alpha, \beta) = O(m(\alpha, \beta))$ *samples.*

*The functions* $m_C$, $m_P$, *and* $m_L$ *are related to one another as:* $m_P(\alpha, \beta, \varepsilon) = m_C(\frac{\alpha}{6}, \frac{\beta}{2})$; $m_L(\alpha, \beta) = m_P(\frac{\alpha}{2}, \frac{\beta}{10}, \varepsilon)$ *for any* $\varepsilon > 0$; *and* $m_C(\alpha, \beta) = m_L(\alpha, \beta)$. *Hence, if there exists a polynomial* $m : (0, 1]^2 \to \mathbb{N}$, *such that* $m_C(\alpha, \beta) = O(m(\alpha, \beta))$, *then* $m_C(\alpha, \beta)$, $m_P(\alpha, \beta)$, *and* $m_L(\alpha, \beta)$ *are all within constant factors of each other.*

## D.1 Compression implies public-private learning

*Proof of Proposition 3.1.* The proof this proposition closely mirrors that of Theorem 4.5 from [ABDH+20]. We adapt their result to the public-private setting.

Fix $\alpha, \beta \in (0, 1]$ and $\varepsilon > 0$. Let $\tau = \tau(\frac{\alpha}{6}, \frac{\beta}{2})$, $t = t(\frac{\alpha}{6}, \frac{\beta}{2})$, and $m = m_C(\frac{\alpha}{6}, \frac{\beta}{2})$. We draw a public dataset $\tilde{X}$ of size $m$ i.i.d. from $p$. Consider

$$\mathcal{S} := \left\{ (S', b) : S' \subseteq \tilde{X} \text{ where } |S'| = \tau, \text{ and } b \in \{0, 1\}^t \right\}.$$

Note that the encoding $f_p(\tilde{X}) \in \mathcal{S}$, so forming $\widehat{\mathcal{Q}} = \{g(S', b) : (S', b) \in \mathcal{S}\}$ means that with probability $\geq 1 - \frac{\beta}{2}$ over the sampling of $\tilde{X}$, $q = g(f_p(\tilde{X})) \in \widehat{\mathcal{Q}}$ has $\text{TV}(q, p) \leq \frac{\alpha}{6}$.

Now, we run the $\varepsilon$-DP 3-agnostic learner from Fact C.2 on $\widehat{\mathcal{Q}}$, targeting error $\frac{\alpha}{2}$ and failure probability $\frac{\beta}{2}$, which is achieved as long as we have $n$ private samples (given in the statement of Proposition 3.1), which is logarithmic in $|\mathcal{S}|$. With probability $\geq 1 - \beta$, we approximately recover $p$ with the compression scheme and the DP learner succeeds, and so the output $Q$ satisfies

$$\text{TV}(Q, p) \leq 3 \cdot \min_{q \in \widehat{\mathcal{Q}}} \text{TV}(p, q) + \frac{\alpha}{2}$$
$$\leq 3 \cdot \frac{\alpha}{6} + \frac{\alpha}{2} = \alpha. \qquad \square$$

## D.2 List learning implies sample compression

*Proof of Proposition 3.5.* Fix any $\alpha, \beta \in (0, 1]$. Let $m = m_L(\alpha, \beta)$ and $\ell = \ell(\alpha, \beta)$. By assumption, $\mathcal{Q}$ admits an $(\alpha, \beta, \ell)$-list learner $\mathcal{L} : \mathcal{X}^m \to \{L \subseteq \Delta(\mathcal{X}) : |L| \leq \ell\}$ that takes $m$ samples. Letting $\tau = m$ and $t = \log_2(\ell)$, we define the compression scheme as follows.

- Encoder: for any $q \in \mathcal{Q}$, the encoder $f_q : \mathcal{X}^m \to \mathcal{X}^\tau \times \{0, 1\}^t$ produces the following, given an input $\tilde{x} \in \mathcal{X}^m$. It first runs the list learner on $\tilde{x}$, obtaining $\mathcal{L}(\tilde{x})$. Then, it finds the smallest index $i$ with $\text{TV}(q, \mathcal{L}(\tilde{x})_i) = \text{dist}(q, \mathcal{L}(\tilde{x}))$, where $\mathcal{L}(\tilde{x})_i$ denotes the $i$-th element of the the list $\mathcal{L}(\tilde{x})$. The output of the list learner is $(\tilde{x}, i)$. Note that $\tilde{x} \in \mathcal{X}^\tau$ and that $i$ can be represented with $\log_2(\ell) = t$ bits.

- Decoder: the fixed decoder $g : \mathcal{X}^\tau \times \{0, 1\}^t \to \Delta(\mathcal{X})$ takes $\tilde{x}$ and $i$, runs the list learner $\mathcal{L}$ on $\tilde{x}$, and produces $\mathcal{L}(\tilde{x})_i$.

By the guarantee of the list learner, we indeed have for any $q \in \mathcal{Q}$, with probability $\geq 1 - \beta$ over the sampling of $\tilde{X} \sim q^m$, $\text{TV}(q, g(f_q(S))) \leq \alpha$. $\qquad \square$

# E   Statements and proofs for Section 4 – Applications

## E.1   Public-private learnability of Gaussians and mixtures of Gaussians

**Corollary E.1** (Public-private learning for Gaussians). *Let $d \geq 1$. The class of Gaussians over $\mathbb{R}^d$ is public-privately learnable with $m(\alpha, \beta, \varepsilon)$ public samples and $n(\alpha, \beta, \varepsilon)$ private samples, where*

$$m(\alpha, \beta, \varepsilon) = O\left(d \log\left(\frac{1}{\beta}\right)\right),$$

$$n(\alpha, \beta, \varepsilon) = O\left(\frac{d^2 \log\left(\frac{d}{\alpha}\right) + \log\left(\frac{1}{\beta}\right)}{\alpha^2} + \frac{d^2 \log\left(\frac{d}{\alpha}\right) + \log\left(\frac{1}{\beta}\right)}{\alpha \varepsilon}\right).$$

**Corollary E.2** (Public-private learning for mixtures of Gaussians). *Let $d, k \geq 1$. The class of all $k$-mixtures of Gaussians over $\mathbb{R}^d$ is public-privately learnable with $m(\alpha, \beta, \varepsilon)$ public samples and $n(\alpha, \beta, \varepsilon)$ private samples, where*

$$m(\alpha, \beta, \varepsilon) = O\left(\frac{kd \log\left(\frac{k}{\beta}\right) \log\left(\frac{1}{\beta}\right)}{\alpha}\right),$$

$$n(\alpha, \beta, \varepsilon) = O\left(\left(\frac{1}{\alpha^2} + \frac{1}{\varepsilon \alpha}\right) \cdot \left(kd^2 \log\left(\frac{d}{\alpha}\right) + kd \log\left(\frac{kd \log\left(\frac{k}{\beta}\right)}{\alpha}\right) + \log\left(\frac{1}{\beta}\right)\right)\right).$$

## E.2   Public-private learnability of mixture and product distributions

**Mixture distributions.**   We first mention a fact from [ABDH$^+$20], which says that if a compression scheme exists for a class of distributions $\mathcal{Q}$, then there exists a compression scheme for the class of $k$-mixtures of $\mathcal{Q}$.

**Fact E.3** (Compression for mixture distributions [ABDH$^+$20, Lemma 4.8]). *If a class of distributions $\mathcal{Q}$ admits $(\tau(\alpha, \beta), t(\alpha, \beta), m(\alpha, \beta))$ realizable sample compression, then for any $k \geq 1$, the class of $k$-mixtures of $\mathcal{Q}$ admits $(\tau_k(\alpha, \beta), t_k(\alpha, \beta), m_k(\alpha, \beta))$ realizable sample compression, where $\tau_k, t_k, m_k : (0, 1]^2 \to \mathbb{N}$ are as follows:*

$$\tau_k(\alpha, \beta) = k \cdot \tau\left(\frac{\alpha}{3}, \beta\right), \quad t_k(\alpha, \beta) = k \cdot t\left(\frac{\alpha}{3}, \beta\right) + \log_2\left(\frac{3k}{\alpha}\right),$$

$$m_k(\alpha, \beta) = \frac{48k \log\left(\frac{6k}{\beta}\right)}{\alpha} \cdot m\left(\frac{\alpha}{3}, \beta\right).$$

Next, we state a corollary of Propositions 3.4 and 3.5, which describes the existence of a compression scheme, given the existence of a public-private learner.

**Corollary E.4** (Public-private learning $\Longrightarrow$ compression). *Let $\mathcal{Q} \subseteq \Delta(\mathcal{X})$ be a class of distributions. Suppose $\mathcal{Q}$ is public-privately learnable with $m_P(\alpha, \beta, \varepsilon)$ public samples and $n(\alpha, \beta, \varepsilon)$ private samples. Then for any $\varepsilon > 0$, $\mathcal{Q}$ admits*

$$(\tau(\alpha, \beta), t(\alpha, \beta), m(\alpha, \beta)) = \left(m_P\left(\frac{\alpha}{2}, \frac{\beta}{10}, \varepsilon\right), \frac{\log(\frac{10}{9}) + \varepsilon \cdot n\left(\frac{\alpha}{2}, \frac{\beta}{10}, \varepsilon\right)}{\log(2)}, m_P\left(\frac{\alpha}{2}, \frac{\beta}{10}, \varepsilon\right)\right)$$

*realizable sample compression.*

*Proof.* Fix $\varepsilon > 0$. From Proposition 3.4, if $\mathcal{Q}$ is public-privately learnable, then it is list learnable to list size $\ell(\alpha, \beta)$ with $m_L(\alpha, \beta)$ samples, where

$$\ell(\alpha, \beta) = \frac{10}{9} \exp\left(\varepsilon \cdot n\left(\frac{\alpha}{2}, \frac{\beta}{10}, \varepsilon\right)\right) \quad \text{and} \quad m_L(\alpha, \beta) = m_P\left(\frac{\alpha}{2}, \frac{\beta}{10}, \varepsilon\right).$$

Proposition 3.5 implies $\mathcal{Q}$ admits $(\tau(\alpha, \beta), t(\alpha, \beta), m_C(\alpha, \beta))$ sample compression, where

$$\tau(\alpha, \beta) = m_L(\alpha, \beta) = m_P\left(\frac{\alpha}{2}, \frac{\beta}{10}, \varepsilon\right),$$

$$t(\alpha, \beta) = \log_2(\ell(\alpha, \beta)) = \frac{\log(\frac{10}{9}) + \varepsilon \cdot n\left(\frac{\alpha}{2}, \frac{\beta}{10}, \varepsilon\right)}{\log(2)},$$

$$m_C(\alpha, \beta) = m_L(\alpha, \beta) = m_P\left(\frac{\alpha}{2}, \frac{\beta}{10}, \varepsilon\right).$$

This completes the proof. $\qquad\square$

As a consequence of Corollary E.4, Fact E.3, and Proposition 3.1, we have the following result about the public-private learnability of mixture distributions.

**Theorem E.5** (Public-private learning for mixture distributions)**.** *Suppose $\mathcal{Q} \subseteq \Delta(\mathcal{X})$ is public-privately learnable with $m(\alpha, \beta, \varepsilon)$ public samples and $n(\alpha, \beta, \varepsilon)$ private samples. Then for any $k \geq 1$, $\mathcal{Q}^{\oplus k}$, the class of $k$-mixtures of $\mathcal{Q}$, is public-privately learnable with $m_k(\alpha, \beta, \varepsilon)$ public samples and $n_k(\alpha, \beta, \varepsilon)$ private samples, where*

$$m_k(\alpha, \beta, \varepsilon) = O\left(\frac{k \log\left(\frac{k}{\beta}\right)}{\alpha} \cdot m\left(\frac{\alpha}{36}, \frac{\beta}{20}, \varepsilon_0\right)\right),$$

$$n_k(\alpha, \beta, \varepsilon) = O\left(\left(\frac{1}{\alpha^2} + \frac{1}{\varepsilon\alpha}\right) \cdot \left(\varepsilon_0 k \cdot n\left(\frac{\alpha}{36}, \frac{\beta}{20}, \varepsilon_0\right) + \right.\right.$$
$$\left.\left. k \log\left(\frac{k \log\left(\frac{k}{\beta}\right)}{\alpha} \cdot m\left(\frac{\alpha}{36}, \frac{\beta}{20}, \varepsilon_0\right)\right) \cdot m\left(\frac{\alpha}{36}, \frac{\beta}{20}, \varepsilon_0\right) + \log\left(\frac{1}{\beta}\right)\right)\right)$$

*for any choice of $\varepsilon_0 > 0$.*

We give an example of an application of this result. Consider the class of Gaussians over $\mathbb{R}^d$, for which there exists a public-private learner that uses $m = O(d)$ public samples and $n = O\left(\frac{d^2}{\alpha^2} + \frac{d^2}{\varepsilon\alpha}\right) \cdot$ polylog $\left(d, \frac{1}{\alpha}, \frac{1}{\beta}\right)$ private samples [BKS22]. Then Theorem E.5 implies that there exists a public-private learner for the class of $k$-mixtures of Gaussians that uses $m_k = O\left(\frac{kd \log(k/\beta)}{\alpha}\right)$ public samples and

$$n_k = O\left(\left(\frac{1}{\alpha^2} + \frac{1}{\varepsilon\alpha}\right) \cdot \left(\varepsilon_0 k \left(\frac{d^2}{\alpha^2} + \frac{d^2}{\varepsilon_0\alpha}\right) + kd\right)\right) \cdot \text{polylog}\left(d, k, \frac{1}{\alpha}, \frac{1}{\beta}\right)$$

private samples for any $\varepsilon_0 > 0$.

With the choice of $\varepsilon_0 = \alpha$, we get a private sample complexity of $n_k = O\left(\frac{kd^2}{\alpha^3} + \frac{kd^2}{\alpha^2\varepsilon}\right) \cdot$ polylog $\left(d, k, \frac{1}{\alpha}, \frac{1}{\beta}\right)$. Notably, this private sample complexity, obtained by specializing the general result of Theorem E.5, suffers some loss compared to our learner for mixtures of Gaussians from Corollary E.2.

**Product distributions.** We start by mentioning a fact from [ABDH+20], which says that if a compression scheme exists for a class of distributions $\mathcal{Q}$, then there exists a compression scheme for the class of $k$-products of $\mathcal{Q}$.

**Fact E.6** (Compression for product distributions [ABDH+20, Lemma 4.6])**.** *If a class of distributions $\mathcal{Q}$ admits $(\tau(\alpha, \beta), t(\alpha, \beta), m(\alpha, \beta))$ $r$-robust sample compression, then for any $k \geq 1$, the class of $k$-products of $\mathcal{Q}$ admits $(\tau_k(\alpha, \beta), t_k(\alpha, \beta), m_k(\alpha, \beta))$ $r$-robust sample compression, where $\tau_k, t_k, m_k : (0, 1] \to \mathbb{N}$ are as follows:*

$$\tau_k(\alpha, \beta) = k \cdot \tau\left(\frac{\alpha}{k}, \beta\right), \quad t_k(\alpha, \beta) = k \cdot t\left(\frac{\alpha}{k}, \beta\right), \quad m_k(\alpha, \beta) = \log_3\left(\frac{3k}{\beta}\right) \cdot m\left(\frac{\alpha}{k}, \beta\right).$$

As a consequence of Corollary E.4, Fact E.6, and Proposition 3.1, we have the following result about the public-private learnability of product distributions.

**Theorem E.7** (Public-private learning for mixture distributions). *Suppose $\mathcal{Q} \subseteq \Delta(\mathcal{X})$ is public-privately learnable with $m(\alpha, \beta, \varepsilon)$ public samples and $n(\alpha, \beta, \varepsilon)$ private samples. Then for any $k \geq 1$, $\mathcal{Q}^{\otimes k}$, the class of $k$-products of $\mathcal{Q}$ over $\mathcal{X}^k$, is public-privately learnable with $m_k(\alpha, \beta, \varepsilon)$ public samples and $n_k(\alpha, \beta, \varepsilon)$ private samples, where*

$$m_k(\alpha, \beta, \varepsilon) = O\left(\log\left(\frac{k}{\beta}\right) \cdot m\left(\frac{\alpha}{12k}, \frac{\beta}{20}, \varepsilon_0\right)\right),$$

$$n_k(\alpha, \beta, \varepsilon) = O\left(\left(\frac{1}{\alpha^2} + \frac{1}{\varepsilon\alpha}\right) \cdot \left(\varepsilon_0 k \cdot n\left(\frac{\alpha}{12k}, \frac{\beta}{20}, \varepsilon_0\right) + \right.\right.$$
$$\left.\left. k \log\left(\log\left(\frac{k}{\beta}\right) \cdot m\left(\frac{\alpha}{12k}, \frac{\beta}{20}, \varepsilon_0\right)\right) \cdot m\left(\frac{\alpha}{12k}, \frac{\beta}{20}, \varepsilon_0\right) + \log\left(\frac{1}{\beta}\right)\right)\right)$$

*for any choice of $\varepsilon_0 > 0$.*

As an example, for the class of Gaussians over $\mathbb{R}$, there exists a public-private learner that requires $m = O(1)$ public samples and $n = O\left(\frac{1}{\alpha^2} + \frac{1}{\varepsilon\alpha}\right) \cdot \text{polylog}\left(\frac{1}{\alpha}, \frac{1}{\beta}\right)$ private samples [BKS22]. Then Theorem E.7 implies that there exists a public-private learner for the class of $k$-products of Gaussians that requires $m_k = O\left(\log(k/\beta)\right)$ public samples and $n_k = \left(\left(\frac{1}{\alpha^2} + \frac{1}{\varepsilon\alpha}\right) \cdot \left(\varepsilon_0 k \left(\frac{1}{\alpha^2} + \frac{1}{\varepsilon_0\alpha}\right) + k\right)\right) \cdot$ polylog $\left(k, \frac{1}{\alpha}, \frac{1}{\beta}\right)$ private samples, for any choice of $\varepsilon_0 > 0$. Note that if were to apply Fact E.6 to Fact 4.1 after setting $d = 1$ in the latter, and then apply Proposition 3.1, we would obtain a better sample complexity in terms of the private data than what we would after combining Corollary E.1 (setting $d = 1$) and Theorem E.7 here. However, Theorem E.7 is a more versatile framework, so some loss is to be expected again.

# F    Statements and proofs for Section 5 – Agnostic and distribution-shifted public-private learning

Theorem 5.2 gives us an agnostic and a distribution-shifted learner for Gaussians over $\mathbb{R}^d$, as stated in the following corollary.

**Corollary F.1** (Agnostic and distribution-shifted public-private learner for Gaussians). *Let $d \geq 1$. For any $\alpha, \beta \in (0, 1]$ and $\varepsilon > 0$, there exists $\frac{1}{3}$-shifted 3-agnostic public-private learner for the class of Gaussians in $\mathbb{R}^d$ that uses $m$ public samples and $n$ private samples, where*

$$m = O\left(d \log\left(\frac{1}{\beta}\right)\right),$$

$$n = O\left(\frac{d^2 \log\left(\frac{d}{\alpha}\right) + \log\left(\frac{1}{\beta}\right)}{\alpha^2} + \frac{d^2 \log\left(\frac{d}{\alpha}\right) + \log\left(\frac{1}{\beta}\right)}{\alpha\varepsilon}\right).$$

# G    Statements and proofs for Section 6 – Lower bounds

**Lemma G.1.** *Let $\mathcal{Q} \subseteq \Delta(\mathcal{X})$ and $m \in \mathbb{N}$. For a subclass $\mathcal{C} \subseteq \mathcal{Q}$, denote by $\mathcal{U}(\mathcal{C})$ the uniform distribution over $\mathcal{C}$. Suppose there exists a sequence of distribution classes $(\mathcal{Q}_k)_{k=1}^{\infty}$, with each $\mathcal{Q}_k \subseteq \mathcal{Q}$, and a set $B \subseteq \mathcal{X}^m$ such that following holds:*

*1. There exists $\eta \in (0, 1]$ and $k_\eta \in \mathbb{N}$ with*

$$\mathbb{P}_{\substack{Q \sim \mathcal{U}(\mathcal{Q}_k) \\ \boldsymbol{X} \sim Q^m}} \{\boldsymbol{X} \in B\} \geq \eta$$

*for all $k \geq k_\eta$.*

2. *There exist $c > 0$ and $\alpha \in (0, 1]$ such that, defining $(u_k)_{k=1}^\infty$, $(r_k)_{k=1}^\infty$, and $(s_k)_{k=1}^\infty$ as*

$$u_k := \sup_{\substack{\boldsymbol{x} \in B \\ q \in \mathcal{Q}_k}} q^m(\boldsymbol{x}),$$

$$r_k := \sup_{p \in \mathcal{Q}_k} \mathbb{P}_{Q \sim \mathcal{U}(\mathcal{Q}_k)} \{\mathrm{TV}(p, Q) \leq 2\alpha\},$$

$$s_k := \inf_{\boldsymbol{x} \in B} \mathbb{P}_{Q \sim \mathcal{U}(\mathcal{Q}_k)} \{Q^m(\boldsymbol{x}) \geq c \cdot u_k\},$$

*we have that*

$$\lim_{k \to \infty} \frac{r_k}{s_k} = 0.$$

*Then for any $\ell \in \mathbb{N}$, there does not exist any $(\frac{\alpha \eta}{4}, \frac{\alpha \eta}{4}, \ell)$-list learner for $\mathcal{Q}$ that uses $m$ samples.*

*Proof.* We provide a proof by contradiction. Suppose for some $\ell \in \mathbb{N}$, we have an $(\frac{\alpha \eta}{4}, \frac{\alpha \eta}{4}, \ell)$-list learner for $\mathcal{Q}$ using $m$ samples, denoted by $\mathcal{L} : \mathcal{X}^m \to \{L \subseteq \Delta(\mathcal{X}) : |L| \leq \ell\}$. Then for all $k \in \mathbb{N}$, we have that

$$\mathbb{E}_{\substack{Q \sim \mathcal{U}(\mathcal{Q}_k) \\ \boldsymbol{X} \sim Q^m}} [\mathrm{dist}(Q, \mathcal{L}(\boldsymbol{X}))] \leq (1 - \tfrac{\alpha \eta}{4}) \cdot \tfrac{\alpha \eta}{4} + \tfrac{\alpha \eta}{4} \cdot 1 \leq \tfrac{\alpha \eta}{2}. \tag{1}$$

Now, since $\lim_{k \to \infty} \frac{r_k}{s_k} = 0$, there exists $k_0 \geq k_\eta \in \mathbb{N}$, such that

$$\frac{r_{k_0} \cdot u_{k_0} \cdot \ell}{s_{k_0} \cdot c u_{k_0}} \leq \frac{1}{11}, \tag{2}$$

and

$$\mathbb{P}_{\substack{Q \sim \mathcal{U}(\mathcal{Q}_{k_0}) \\ \boldsymbol{X} \sim Q^m}} \{\boldsymbol{X} \in B\} \geq \eta. \tag{3}$$

Fix any $\boldsymbol{x} \in B$, and let $R = \{q \in \mathcal{Q}_{k_0} : \mathrm{dist}(q, \mathcal{L}(\boldsymbol{x})) \leq \alpha\}$ and $S = \{q \in \mathcal{Q}_{k_0} : q^m(\boldsymbol{x}) \geq c u_{k_0}\}$ (uote that both $R$ and $S$ depend on $\boldsymbol{x}$).

For $i \in [\ell]$, further let $R_i = \{q \in \mathcal{Q}_{k_0} : \mathrm{TV}(q, \mathcal{L}(\boldsymbol{x})_i) \leq \alpha\}$, so that $R = \cup_{i=1}^\ell R_i$.

Now, fix $i \in [\ell]$. Assuming that $R_i \neq \emptyset$, consider any $p \in R_i$. For any $q \in R_i$, we have $\mathrm{TV}(p, q) \leq 2\alpha$. Hence, $R_i \subseteq \{q \in \mathcal{Q}_{k_0} : \mathrm{TV}(p, q) \leq 2\alpha\}$. Regardless of whether $R_i$ is empty,

$$\mathbb{P}_{Q \sim \mathcal{U}(\mathcal{Q}_{k_0})} \{Q \in R_i\} \leq \sup_{p \in \mathcal{Q}_{k_0}} \mathbb{P}_{Q \sim \mathcal{U}(\mathcal{Q}_{k_0})} \{\mathrm{TV}(p, Q) \leq 2\alpha\} = r_{k_0}.$$

Moreover, we can conclude that

$$\mathbb{P}_{Q \sim \mathcal{U}(\mathcal{Q}_{k_0})} \{Q \in R\} \leq \sum_{i=1}^\ell \mathbb{P}_{Q \sim \mathcal{U}(\mathcal{Q}_{k_0})} \{Q \in R_i\} \leq r_{k_0} \cdot \ell. \tag{4}$$

Observe that this implies, since $u_{k_0} \geq q^m(\boldsymbol{x})$,

$$\int_R q^m(\boldsymbol{x}) f_Q(q) dq \leq u_{k_0} \int_R f_Q(q) dq = u_{k_0} \mathbb{P}_{Q \sim \mathcal{U}(\mathcal{Q}_{k_0})} \{Q \in R\} \leq u_{k_0} \cdot r_{k_0} \cdot \ell \tag{5}$$

an inequality we will use momentarily. We can now write

$$\underset{\substack{Q\sim\mathcal{U}(\mathcal{Q}_{k_0})\\ \boldsymbol{X}\sim Q^m}}{\mathbb{E}}[\operatorname{dist}(Q,\mathcal{L}(\boldsymbol{X}))\mid \boldsymbol{X}=\boldsymbol{x}] = \int_{\mathcal{Q}_{k_0}} f_{Q\mid\boldsymbol{X}}(q\mid\boldsymbol{x})\cdot\operatorname{dist}(q,\mathcal{L}(\boldsymbol{x}))dq$$

$$\geq \int_{S\setminus R} f_{Q\mid\boldsymbol{X}}(q\mid\boldsymbol{x})\cdot\operatorname{dist}(q,\mathcal{L}(\boldsymbol{x}))dq$$

$$\geq \alpha\int_{S\setminus R} f_{Q\mid\boldsymbol{X}}(q\mid\boldsymbol{x})dq$$

$$\geq \alpha\left(\int_S f_{Q\mid\boldsymbol{X}}(q\mid\boldsymbol{x})dq - \int_R f_{Q\mid\boldsymbol{X}}(q\mid\boldsymbol{x})dq\right)$$

$$= \alpha\left(\int_S \frac{q^m(\boldsymbol{x})f_Q(q)}{f_X(\boldsymbol{x})}dq - \int_R \frac{q^m(\boldsymbol{x})f_Q(q)}{f_X(\boldsymbol{x})}dq\right)$$

$$= \alpha\frac{1}{f_X(\boldsymbol{x})}\left(\int_S q^m(\boldsymbol{x})f_Q(q)dq - \int_R q^m(\boldsymbol{x})f_Q(q)dq\right)$$

$$\geq \alpha\frac{1}{f_X(\boldsymbol{x})}\left(cu_{k_0}\int_S f_Q(q)dq - u_{k_0}\cdot\ell\cdot r_{k_0}\right)$$

$$\text{(By definition of } S \text{ and (5))}$$

$$= \alpha\frac{1}{f_X(\boldsymbol{x})}\left(cu_{k_0}\underset{Q\sim\mathcal{U}(\mathcal{Q}_{k_0})}{\mathbb{P}}\{Q\in S\} - u_{k_0}\cdot\ell\cdot r_{k_0}\right).$$

Plugging (4) in, along with the definition of $s_{k_0}$, we have,

$$\underset{\substack{Q\sim\mathcal{U}(\mathcal{Q}_{k_0})\\ \boldsymbol{X}\sim Q^m}}{\mathbb{E}}[\operatorname{dist}(Q,\mathcal{L}(\boldsymbol{X}))\mid \boldsymbol{X}=\boldsymbol{x}] \geq \alpha\frac{1}{f_X(\boldsymbol{x})}\left(cu_{k_0}\cdot s_{k_0} - u_{k_0}\cdot\ell\cdot r_{k_0}\right)$$

$$\geq \alpha\frac{1}{f_X(\boldsymbol{x})}(10\cdot u_{k_0}\cdot\ell\cdot r_{k_0}) \qquad (k_0 \text{ from Equation 2})$$

$$\geq 10\alpha\int_R \frac{q^m(\boldsymbol{x})f_Q(q)}{f_X(\boldsymbol{x})}dq \qquad\qquad \text{(by (5))}$$

$$= 10\alpha\cdot\underset{\substack{Q\sim\mathcal{U}(\mathcal{Q}_{k_0})\\ X\sim Q^m}}{\mathbb{P}}\{Q\in R\mid \boldsymbol{X}=\boldsymbol{x}\}.$$

Integrating over all $\boldsymbol{x}\in B$ and using Inequality 3,

$$\underset{\substack{Q\sim\mathcal{U}(\mathcal{Q}_{k_0})\\ \boldsymbol{X}\sim Q^m}}{\mathbb{E}}[\operatorname{dist}(Q,\mathcal{L}(\boldsymbol{X}))] \geq \underset{\substack{Q\sim\mathcal{U}(\mathcal{Q}_{k_0})\\ \boldsymbol{X}\sim Q^m}}{\mathbb{P}}\{\boldsymbol{X}\in B\}\cdot\underset{\substack{Q\sim\mathcal{U}(\mathcal{Q}_{k_0})\\ \boldsymbol{X}\sim Q^m}}{\mathbb{E}}[\operatorname{dist}(Q,\mathcal{L}(\boldsymbol{X}))\mid \boldsymbol{X}\in B]$$

$$\geq \eta\cdot\underset{\substack{Q\sim\mathcal{U}(\mathcal{Q}_{k_0})\\ \boldsymbol{X}\sim Q^m}}{\mathbb{E}}[\operatorname{dist}(Q,\mathcal{L}(\boldsymbol{X}))\mid \boldsymbol{X}\in B]$$

$$\geq \eta\cdot 10\alpha\cdot\underset{\substack{Q\sim\mathcal{U}(\mathcal{Q}_{k_0})\\ \boldsymbol{X}\sim Q^m}}{\mathbb{P}}\{\operatorname{dist}(Q,\mathcal{L}(\boldsymbol{X}))\leq\alpha\mid \boldsymbol{X}\in B\}.$$

If $\mathbb{P}\{\operatorname{dist}(Q,\mathcal{L}(\boldsymbol{X}))\leq\alpha\mid \boldsymbol{X}\in B\}\geq\frac{1}{10}$, then $\mathbb{E}[\operatorname{dist}(Q,\mathcal{L}(\boldsymbol{X}))]\geq\alpha\eta$, contradicting (1). Otherwise,

$$\mathbb{E}[\operatorname{dist}(Q,\mathcal{L}(\boldsymbol{X}))] \geq \eta\cdot\mathbb{E}[\operatorname{dist}(Q,\mathcal{L}(\boldsymbol{X}))\mid \boldsymbol{X}\in B]$$

$$\geq \eta\cdot\alpha\cdot\mathbb{P}\{\operatorname{dist}(Q,\mathcal{L}(\boldsymbol{X}))>\alpha\mid \boldsymbol{X}\in B\}$$

$$\geq \eta\cdot(\alpha\cdot(1-\tfrac{1}{10})),$$

also contradicting Equation 1. $\qquad\qquad\square$

*Proof of Theorem 6.1.* To prove Theorem 6.1, it suffices to find, for every $d\geq 2$, a sequence of subclasses $(\mathcal{Q}_k)_{k=1}^\infty$ and a set $B\in(\mathbb{R}^d)^{d-1}$ that indeed satisfy the conditions of Lemma G.1. In what follows, we fix an arbitrary $d\geq 2$.

**The construction of the sequence of hard subclasses.** Let $e_d = [0, 0, \ldots, 1]^\top \in \mathbb{R}^d$. We define the following sets:

$$T = \left\{ \begin{bmatrix} t \\ 0 \end{bmatrix} \in \mathbb{R}^d : t \in \mathbb{R}^{d-1} \text{ with } \|t\|_2 \leq \frac{1}{2} \right\},$$

$$C = \left\{ \begin{bmatrix} t \\ \lambda \end{bmatrix} \in \mathbb{R}^d : t \in \mathbb{R}^{d-1} \text{ with } \|t\|_2 \leq \frac{1}{2} \text{ and } \lambda \in [1, 2] \subseteq \mathbb{R} \right\}.$$

That is, $T$ is a $\frac{1}{2}$-disk (a disk with radius $\frac{1}{2}$) in $\mathbb{R}^{d-1}$ embedded onto the $(d-1)$-dimensional hyperplane in $\mathbb{R}^d$ spanning the first $(d-1)$ dimensions (axes), centered at the origin. $C$ is a cylinder of unit length and radius $\frac{1}{2}$ placed unit distance away from $T$ in the positive $e_d$-direction.

Let $S^{d-1} = \{x \in \mathbb{R}^d : \|x\|_2 = 1\}$ be the unit-sphere, centered at the origin, in $\mathbb{R}^d$, and let

$$N = \left\{ u \in S^{d-1} : |u \cdot e_d| \leq \frac{\sqrt{3}}{2} \right\}.$$

That is, $N$ is the set of vectors $u$ on the unit hypersphere with angle $\geq \frac{\pi}{6}$ from $e_d$. For $u \in N$, define the "rotatiou" matrix

$$R_u = \begin{bmatrix} | & | & & | \\ u & v_2 & \ldots & v_d \\ | & | & & | \end{bmatrix} \in \mathbb{R}^{d \times d}$$

where $\{v_2, \ldots, v_d\}$ is any orthonormal basis for $\{u\}^\perp$ (where $\{u\}^\perp$ denotes the subspace orthogonal to the subspace spanned by the set of vectors $\{u\}$).[4]

Now, for $\sigma > 0$, $t \in T$, and $u \in N$, define the Gaussian

$$G(\sigma, t, u) = \mathcal{N}\left( t, R_u \begin{bmatrix} \sigma^2 & & & \\ & 1 & & O \\ & & \ddots & \\ & O & & 1 \end{bmatrix} R_u^\top \right) \in \Delta(\mathbb{R}^d).$$

For all $k \geq 1$, let

$$Q_k = \left\{ G\left( \frac{1}{k}, t, u \right) : t \in T, u \in N \right\}.$$

That is, each $Q_k$ is a class of "flat" (i.e., near $(d-1)$-dimensional) Gaussians in $\mathbb{R}^d$, with $\sigma^2 = \frac{1}{k^2}$ variance on a single thin direction $u$ and unit variance in all other directions. Their mean vectors come from a point on the hyperplanar disk $T$ (which we recall is a $(d-1)$-dimensional disk orthogonal to $e_d$), and the thin direction $u$ comes from $N$ (which is $S^{d-1}$ excluding points that form angle $< \frac{\pi}{6}$ with $e_d$). As $k \to \infty$, the Gaussians get flatter.

**Lower bounding the weight of $B$.** We start with the following claim, which shows the probability that $d-1$ samples drawn the uniform mixture of $Q_k^{d-1}$ all fall into the cylinder $C$ can be uniformly lower bounded by an absolute constant, independent of $k$.

**Claim G.2.** *Let $B$ be the set of all possible vectors of $d-1$ points in the cylinder $C$, i..e, $B = C^{d-1} \in (\mathbb{R}^d)^{d-1}$. There exists $\eta > 0$ such that for $k \geq 10$,*

$$\mathop{\mathbb{P}}_{\substack{Q \sim \mathcal{U}(Q_k) \\ \boldsymbol{X} \sim Q^{d-1}}} \{\boldsymbol{X} \in B\} \geq \eta.$$

*Proof of Claim G.2.* Consider the inscribed cylinder $C' \subseteq C$

$$C' = \left\{ \begin{bmatrix} t \\ \lambda \end{bmatrix} \in \mathbb{R}^d : t \in \mathbb{R}^{d-1} \text{ with } \|t\|_2 \leq \frac{1}{3} \text{ and } \lambda \in \left[ \frac{4}{3}, \frac{5}{3} \right] \subseteq \mathbb{R} \right\}.$$

---

[4]Technically, $R_u$ is an equivalence class of matrices since we do not specify which orthonormal basis of $\{u\}^\perp$. However, as it turns out, the choice of the orthonormal basis of $\{u\}^\perp$ does not matter since they all result in the same Gaussian densities in the proceeding definition of $G(\sigma, t, u)$.

Also, consider $T' \subseteq T$ and $N' \subseteq N$:

$$T' = \left\{ \begin{bmatrix} t \\ 0 \end{bmatrix} \in \mathbb{R}^d : t \in \mathbb{R}^{d-1} \text{ with } \|t\|_2 \leq \frac{1}{4} \right\},$$

$$N' = \left\{ u \in S^{d-1} : |u \cdot e_d| \leq \frac{1}{36} \right\}.$$

Now, fix $u \in N'$ and $t \in T'$. Define the plane going through $t$ with normal vector $u$ as,

$$P(u,t) = \left\{ t + x : x \in \mathbb{R}^d \text{ with } x \cdot u = 0 \right\}.$$

First, we show $P(u,t) \cap C'$ contains a $(d-1)$-dimensional region. Consider,

$$y = \begin{bmatrix} t \\ \frac{3}{2} \end{bmatrix}.$$

The projection onto $P(u,t)$ of $y$ is given by,

$$y' = (y - t) - ((y - t) \cdot u)u + t = \begin{bmatrix} t \\ \frac{3}{2} \end{bmatrix} - cu,$$

where $|c| = |(y - t) \cdot u| \leq \frac{3}{2} \cdot \frac{1}{36} = \frac{1}{24}$. Since $\|t\|_2 \leq \frac{1}{4}$, the norm of the first $(d-1)$ dimensions of $y'$ is $\leq \frac{1}{4} + \frac{1}{24} \leq \frac{1}{3}$ and $y'_d \in [\frac{3}{2} - \frac{1}{24}, \frac{3}{2} + \frac{1}{24}]$, and so $y' \in C'$. Moreover, adding any $z$ with $z \cdot u = 0$ and $\|z\|_2 \leq \frac{1}{24}$ results in $y' + z$ with the norm of the first $d - 1$ dimensions being at most $\frac{1}{4} + \frac{1}{24} + \frac{1}{24} \leq \frac{1}{3}$ and $(y' + z)_d \in [\frac{3}{2} - \frac{1}{12}, \frac{3}{2} + \frac{1}{12}]$. Hence, $y' + z \in C'$. This shows that $P(u,t) \cap C'$ contains a $(d-1)$-dimensional subspace, since it contains a $(d-1)$-dimensional disk of radius $\frac{1}{24}$.

Next, let

$$M = \left\{ p + su : p \in C' \cap P(u,t), s \in \left[ -\frac{1}{6}, \frac{1}{6} \right] \subseteq \mathbb{R} \right\}.$$

That is, $M$ is a rectangular "extrusion" of $C' \cap P(u,t)$ along both its normal vectors. Indeed, we have $M \subseteq C$, since adding a vector of length $\leq \frac{1}{6}$ cannot take a point in $C'$ outside of $C$. We also have that $M$ is a $d$-dimensional region, so

$$\underset{X \sim G(1/10, t, u)}{\mathbb{P}} \{X \in C\} \geq \underset{X \sim G(1/10, t, u)}{\mathbb{P}} \{X \in M\} > 0.$$

Note that for $\sigma \leq \frac{1}{10}$, we have

$$\underset{X \sim G(\sigma, t, u)}{\mathbb{P}} \{X \in M\} \geq \underset{X \sim G(1/10, t, u)}{\mathbb{P}} \{X \in M\}.$$

This is because any $x \in M$ can be written as $t + x + cu$, where $x$ is such that $x \cdot u = 0$, and $|c| \leq \frac{1}{6}$. Plugging in this decomposition of $x$ into the densities of $G(1/10, u, t)$ and $G(\sigma, u, t)$, and simplifying yields the above.

To conclude, for $k \geq 10$, we have

$$\underset{\substack{Q \sim \mathcal{U}(\mathcal{Q}_k) \\ \boldsymbol{X} \sim Q^{d-1}}}{\mathbb{P}} \{\boldsymbol{X} \in C^{d-1}\} = \underset{\substack{t \sim \mathcal{U}(T) \\ u \sim \mathcal{U}(N) \\ \boldsymbol{X} \sim G(1/k, t, u)^{d-1}}}{\mathbb{P}} \{\boldsymbol{X} \in C^{d-1}\}$$

$$= c \int_T \int_N \underset{\boldsymbol{X} \sim G(1/k, t, u)^{d-1}}{\mathbb{P}} \{\boldsymbol{X} \in C^{d-1}\} \, du \, dt$$

$$\geq c \int_{T'} \int_{N'} \underset{\boldsymbol{X} \sim G(1/k, t, u)^{d-1}}{\mathbb{P}} \{\boldsymbol{X} \in C^{d-1}\} \, du \, dt$$

$$= c \int_{T'} \int_{N'} \left( \underset{X \sim G(1/k, t, u)}{\mathbb{P}} \{X \in C\} \right)^{d-1} \, du \, dt$$

$$\geq c \int_{T'} \int_{N'} \left( \underset{X \sim G(1/k, t, u)}{\mathbb{P}} \{X \in M\} \right)^{d-1} \, du \, dt$$

$$\geq c \int_{T'} \int_{N'} \left( \underset{X \sim G(1/10, t, u)}{\mathbb{P}} \{X \in M\} \right)^{d-1} \, du \, dt$$

$$=: \eta > 0, \tag{6}$$

where $c = f_T(t) \cdot f_N(u) > 0$ is the uniform density over $T \times N$. Note that the final integral is non-zero since $T' \times N'$ has non-zero measure in $T \times N$ and that $\underset{X \sim G(1/10, t, u)}{\mathbb{P}} \{X \in M\}$ is indeed non-zero for all $t \in T', u \in N'$. $\qquad\square$

**Upper bounding $r_k$, the weight of $\alpha$-TV balls.** We prove the following.

**Claim G.3.** *For $k \geq 1$, let*

$$r_k := \sup_{p \in \mathcal{Q}_k} \underset{Q \sim \mathcal{U}(\mathcal{Q}_k)}{\mathbb{P}} \left\{ \mathrm{TV}(p, Q) \leq \tfrac{1}{400} \right\}.$$

*Then we have,*

$$r_k = O\left( \frac{1}{k^d} \right) \to 0 \quad as \ k \to \infty.$$

We use the following three facts regarding total variation distance, Gaussians, and the surface area of hyperspherical caps.

**Fact G.4** (Data-processing inequality for TV distance)**.** *Let $p, q \in \Delta(\mathcal{X})$. For any measurable $f \colon \mathcal{X} \to \mathcal{Y}$,*

$$\mathrm{TV}(f(p), f(q)) \leq \mathrm{TV}(p, q),$$

*where for $p \in \Delta(\mathcal{X})$, $f(p)$ denotes the push-forward distribution assigning for all measurable $A \subseteq \mathcal{Y}$, $f(p)(A) = p(f^{-1}(A))$.*

**Fact G.5** (TV Distance between 1-Dimensional Gaussians [DMR18, Theorem 1.3])**.** *Let $\mathcal{N}(\mu_1, \sigma_1^2)$ and $\mathcal{N}(\mu_2, \sigma_2^2)$ be Gaussians over $\mathbb{R}$. Then*

$$\frac{1}{200} \cdot \min \left\{ 1, \max \left\{ \frac{|\sigma_1^2 - \sigma_2^2|}{\sigma_1^2}, \frac{40|\mu_1 - \mu_2|}{\sigma_1} \right\} \right\} \leq \mathrm{TV}\left( \mathcal{N}\left( \mu_1, \sigma_1^2 \right), \mathcal{N}\left( \mu_2, \sigma_2^2 \right) \right).$$

**Fact G.6** (Surface area of hyperspherical caps [Li10])**.** *For $u \in S^{d-1}$ and $\theta \in [0, \frac{\pi}{2}]$, define*

$$C(u, \theta) = \left\{ x \in S^{d-1} : \angle(x, u) \leq \theta \right\}$$

*where for $u, v \in S^{d-1}$, $\angle(u, v) := \cos^{-1}(u \cdot v)$. We have*

$$\mathrm{Area}(C(u, \theta)) = \frac{2\pi^{(d-1)/2}}{\Gamma(\frac{d-1}{2})} \cdot \int_0^\theta \sin^{d-2}(x) dx.$$

*Note that*

$$\mathrm{Area}(S^{d-1}) = \frac{2\pi^{d/2}}{\Gamma(\frac{d}{2})}.$$

*Proof of Claim G.3.* Let $\sigma > 0$. Let $t_1, t_2 \in T$ and $u_1, u_2, \in N$. We will compare the total variation distance of the Gaussians defined by these parameters. Let

$$D_\sigma = \begin{bmatrix} \sigma^2 & & & \\ & 1 & & O \\ & & \ddots & \\ & O & & 1 \end{bmatrix}.$$

By Fact G.4, taking $f \colon \mathbb{R}^d \to \mathbb{R}$ to be $f(x) = u_1^\top (x - t_1)$,

$$\mathrm{TV}(G(\sigma, t_1, u_2), G(\sigma, t_2, u_2)) \geq \mathrm{TV}(\mathcal{N}(u_1^\top (t_1 - t_1), u_1^\top R_{u_1} D_\sigma R_{u_1}^\top u_1), \mathcal{N}(u_1^\top (t_2 - t_1), u_1^\top R_{u_2} D_\sigma R_{u_2}^\top u_1))$$
$$= \mathrm{TV}(\mathcal{N}(0, \sigma^2), \mathcal{N}(u_1 \cdot \Delta t, \sigma^2 \cos^2(\angle(u_1, u_2)) + \sin^2(\angle(u_1, u_2)))),$$

where $\Delta t = t_2 - t_1$. For the last line above, we take $R_{u_2} = [u_2, v_2, \ldots, v_d]$, where $\{v_2, \ldots, v_d\}$ is an orthonormal basis for $\{u_2\}^\perp$. Then the equality in the last line for the variance of the second Gaussian uses,

$$
\begin{aligned}
u_1^\top R_{u_2} D_\sigma R_{u_2}^\top u_1 &= \sigma^2 (u_1 \cdot u_2)^2 + (v_2 \cdot u_2)^2 + \cdots + (v_d \cdot u_2)^2 \\
&= \sigma^2 (u_1 \cdot u_2)^2 + (1 - (u_1 \cdot u_2)^2) \\
&= \sigma^2 \cos^2(\angle(u_1, u_2)) + (1 - \cos^2(\angle(u_1, u_2))) \\
&= \sigma^2 \cos^2(\angle(u_1, u_2)) + \sin^2(\angle(u_1, u_2)),
\end{aligned}
$$

where $R_{u_2}$ being unitary implies that $(u_1 \cdot u_2)^2 + (u_1 \cdot v_2)^2 + \cdots + (u_2 \cdot v_d)^2 = 1$, yielding the second equality in the above.

We show that if $\angle(u_1, u_2) \in [\frac{\sqrt{2}\pi}{2}\sigma, \pi - \frac{\sqrt{2}\pi}{2}\sigma]$, $\mathrm{TV}(G(\sigma, t_1, u_1), G(\sigma, t_2, u_2)) \geq \frac{1}{200}$. First, we consider the case where $\angle(u_1, u_2) \in [\frac{\sqrt{2}\pi}{2}\sigma, \frac{\pi}{2}]$. Using that on $[0, \frac{\pi}{2}]$, we have $\sin(x) \geq \frac{2}{\pi}x$ and $\cos(x) \geq 0$, we get

$$
\sigma^2 \cos^2(\angle(u_1, u_2)) + \sin^2(\angle(u_1, u_2)) \geq \frac{4}{\pi^2}\angle(u_1, u_2)^2 \geq 2\sigma^2. \tag{7}
$$

Therefore,

$$
\frac{\sigma_1^2 - \sigma_2^2}{\sigma_1^2} \leq \frac{\sigma^2 - 2\sigma^2}{\sigma^2} \leq -1,
$$

and by Fact G.5, we can conclude that $\mathrm{TV}(G(\sigma, t_1, u_1), G(\sigma, t_2, u_2)) \geq \frac{1}{200}$. Now, consider the case where $\angle(u_1, u_2) \in [\frac{\pi}{2}, \pi - \frac{\sqrt{2}\pi}{2}]$. Note that in this case, there exists $u_2' = -u_2 \in [\frac{\sqrt{2}\pi}{2}, \frac{\pi}{2}]$ with $G(\sigma, t_2, u_2) = G(\sigma, t_2, u_2')$, bringing us back to the previous case.

Next, note that since $\|u_1\|_2 = 1$ and $|u_1^{(d)}| = |u_1 \cdot e_d| \leq \frac{\sqrt{3}}{2}$ (by the definition of $N$), letting $r = [u_1^{(1)}, \ldots, u_1^{(d-1)}]^\top \in \mathbb{R}^{d-1}$, we have $\|r\|_2 \geq \frac{1}{2}$. Let $\hat{r} = \frac{r}{\|r\|_2}$. We have that if $[\Delta t_1, \ldots, \Delta t_{d-1}]^\top \cdot \hat{r} \geq \frac{1}{20}\sigma$, then,

$$
\begin{aligned}
u_1 \cdot \Delta t &= r \cdot [\Delta t_1, \ldots, \Delta t_{d-1}]^\top \\
&\geq \frac{r}{2\|r\|_2} \cdot [\Delta t_1, \ldots, \Delta t_{d-1}]^\top \\
&\geq \tfrac{1}{2}\hat{r} \cdot [\Delta t_1, \ldots, \Delta t_{d-1}]^\top \\
&\geq \frac{1}{40}\sigma.
\end{aligned}
$$

This implies that

$$
\frac{40(\mu_1 - \mu_2)}{\sigma_1} = \frac{40(-u_1 \cdot \Delta t)}{\sigma} \leq -1,
$$

and by Fact G.5, we can conclude $\mathrm{TV}(G(\sigma, t_1, u_1), G(\sigma, t_2, u_2)) \geq \frac{1}{200}$. Therefore, for any $u \in N$, $t \in T$,

$$
\begin{aligned}
\mathop{\mathbb{P}}_{Q \sim \mathcal{U}(\mathcal{Q}_k)} \left\{ \mathrm{TV}(G(\tfrac{1}{k}, t, u), Q) \leq \frac{1}{400} \right\} &= \mathop{\mathbb{P}}_{\substack{t' \sim \mathcal{U}(T) \\ u' \sim \mathcal{U}(N)}} \left\{ \mathrm{TV}(G(\tfrac{1}{k}, t, u), G(\tfrac{1}{k}, t', u')) \leq \frac{1}{400} \right\} \\
&\leq \mathop{\mathbb{P}}_{\substack{t' \sim \mathcal{U}(T) \\ u' \sim \mathcal{U}(N)}} \left\{ \mathrm{TV}(G(\tfrac{1}{k}, t, u), G(\tfrac{1}{k}, t', u')) < \frac{1}{200} \right\} \\
&\leq \mathop{\mathbb{P}}_{t' \sim \mathcal{U}(T)} \left\{ [\Delta t_1, \ldots, \Delta t_{d-1}]^\top \cdot \hat{r} < \tfrac{1}{20k} \right\} \cdot \\
&\quad \mathop{\mathbb{P}}_{u' \sim \mathcal{U}(N)} \left\{ (\angle(u, u') \in [0, \tfrac{\sqrt{2}\pi}{2k}) \cup (\pi - \tfrac{\sqrt{2}\pi}{2k}, \pi]) \right\}.
\end{aligned}
$$

For the first term, note that the event

$$\left\{[\Delta t_1, \ldots, \Delta t_{d-1}] \cdot \hat{r} < \tfrac{1}{20k}\right\} \subseteq \left\{t' \in \left\{t + \begin{bmatrix} x \\ 0 \end{bmatrix} + \lambda \begin{bmatrix} \hat{r} \\ 0 \end{bmatrix} : \|x\|_2 \le 1, x \cdot \hat{r} = 0, \lambda \le \tfrac{1}{20k}\right\}\right\},$$

which under $\mathcal{U}(T)$, for some $c_d > 0$ depending only on $d$, has probability $\le c_d \cdot \tfrac{1}{20k}$.

For the second term, note that $\angle(u, u') \in [0, \tfrac{\sqrt{2}\pi}{2k}) \cup (\pi - \tfrac{\sqrt{2}\pi}{2k}, \pi]$ means $u' \in C(u, \tfrac{\sqrt{2}\pi}{2k}) \cup C(-u, \tfrac{\sqrt{2}\pi}{2k})$. By Fact G.6, we know that under $\mathcal{U}(N)$, for some $c_d$ depending only on $d$,

$$\mathbb{P}_{u' \sim \mathcal{U}(N)}\left\{u' \in C(u, \tfrac{\sqrt{2}\pi}{2k})\right\} = c_d \cdot \int_0^{\sqrt{2}\pi/2k} \sin^{d-2}(x)dx$$

$$\le c_d \cdot \int_0^{\sqrt{2}\pi/2k} x^{d-2}dx$$

$$= \frac{c_d}{d-1}\left(\frac{\sqrt{2}\pi}{2}\right)^{d-1}\frac{1}{k^{d-1}}.$$

The bound is the same for $C(-u, \tfrac{\sqrt{2}\pi}{2k})$. Plugging these into the above, we can conclude that

$$r_k = \sup_{p \in \mathcal{Q}_k} \mathbb{P}_{Q \sim \mathcal{U}(\mathcal{Q}_k)}\left\{\mathrm{TV}(p, Q) \le \frac{1}{400}\right\} \le O\left(\frac{1}{k^d}\right) \to 0 \quad \text{as } k \to \infty.$$

This proves the claim. $\qquad\square$

**Lower bounding $s_k$, the weight of alternative hypotheses.** First, we note that

$$u_k = \sup_{\substack{\boldsymbol{x} \in B \\ q \in \mathcal{Q}_k}} q^{d-1}(\boldsymbol{x}) = \left(\frac{1}{(2\pi)^{d/2}}k\exp(-\tfrac{1}{2})\right)^{d-1},$$

which is achieved by $G(\tfrac{1}{k}, \mathbf{0}, e_1)$ (where $\mathbf{0} \in \mathbb{R}^d$ is the origin) and $\boldsymbol{x} = (e_d, \ldots, e_d)$. Let

$$c = \frac{\exp(-5)^{d-1}}{\exp(-\tfrac{1}{2})^{d-1}} = \exp\left(\frac{9(d-1)}{2}\right).$$

**Claim G.7.** *For $k \ge 1$, letting $(u_k)_{k=1}^\infty$ and $c$ be defined as above, define*

$$s_k := \inf_{\boldsymbol{x} \in B} \mathbb{P}_{Q \sim \mathcal{U}(\mathcal{Q}_k)}\left\{Q^{d-1}(\boldsymbol{x}) \ge cu_k\right\}.$$

*Then we have,*

$$s_k = \Omega\left(\frac{1}{k^{d-1}}\right) \to 0 \quad \text{as } k \to \infty.$$

*Proof of Claim G.7.* Let $k \ge 1$. Fix any $\boldsymbol{x} = (x_1, \ldots, x_{d-1}) \in B$. For every $t \in T$, there exists $u \in \{x_1 - t, x_2 - t, \ldots, x_{d-1} - t\}^\perp$. We show $\angle(u, e_d) \ge \tfrac{\pi}{4}$. Suppose otherwise, that is, $\angle(u, e_d) < \tfrac{\pi}{4} \implies |u \cdot e_d| = |u^{(d)}| > \tfrac{\sqrt{2}}{2}$. Then,

$$u \cdot (x_1 - t) = u^{(1)}(x_1^{(1)} - t^{(1)}) + \cdots + u^{(d-1)}(x_1^{(d-1)} - t^{(d-1)}) + u^{(d)}x_1^{(d)}.$$

By our assumption on $u^{(d)}$, and by the fact that $x_1 \in C$, we have that $|u^{(d)}x_1^{(d)}| > \tfrac{\sqrt{2}}{2}$. By Cauchy-Schwarz in $\mathbb{R}^{d-1}$, we have that,

$$|u^{(1)}(x_1^{(1)} - t^{(1)}) + \cdots + u^{(d-1)}(x_1^{(d-1)} - t^{(d-1)})|$$

$$\le \|[u^{(1)}, \ldots, u^{(d-1)}]^\top\|_2 \cdot \|[x_1^{(1)}, \ldots, x_1^{(d-1)}]^\top - [t^{(1)}, \ldots, t^{(d-1)}]^\top\|_2$$

$$< \frac{\sqrt{2}}{2} \cdot 1.$$

The last inequality uses that $\|u\|_2 = 1$ and $(u^{(d)})^2 > \frac{1}{2}$, so the norm of the first $(d-1)$ coordinates is $< \frac{\sqrt{2}}{2}$, and also the fact that the first $(d-1)$ coordinates of $x$ and $t$ are in the $\frac{1}{2}$-disk. This inequality, combined with the fact that $|u^{(d)}x_1^{(d)}| > \frac{\sqrt{2}}{2}$ contradicts that that $(x_1 - t) \cdot u = 0$.

Now, for the $t$ and the $u$ from above, consider an arbitrary $u'$ with $\angle(u, u') \le \frac{1}{k}$, and the Gaussian with mean $t$ and normal vector $u'$, $G(\frac{1}{k}, t, u')$. We will show that any such Gaussian assigns high mass to the point $x$, and furthermore that there is a high density of such Gaussians. Note that for $k \ge 5$, $\frac{1}{k} \le \frac{\pi}{3} - \frac{\pi}{4} \implies \angle(u', e_d) \ge \frac{\pi}{3} \implies u' \in N$. We compute the minimum density this Gaussian assigns to $x$. Consider, for $i \in [d-1]$,

$$
\begin{aligned}
(x_i - t)^\top (R_{u'} D_{1/k} R_{u'}^\top)^{-1}(x_i - t) &= \|D_{\sqrt{k}} R_{u'}^\top(x_i - t)\|^2 \\
&= k^2 |u' \cdot (x_i - t)|^2 + |v_2 \cdot (x_i - t)|^2 + \cdots + |v_d \cdot (x_i - t)|^2 \\
&\le 5(k^2|u' \cdot \hat{r}|^2 + 1),
\end{aligned}
$$

where $\hat{r} = (x_i - t)/\|x_i - t\|$ and $\{v_2, \ldots, v_d\}$ is an orthonormal basis of $\{u'\}^\perp$. We have that,

$$
\begin{aligned}
|u' \cdot \hat{r}|^2 &= |(u + (u - u')) \cdot \hat{r}|^2 \\
&\le \|u - u'\|_2^2 \cdot \|\hat{r}\|_2^2 \\
&= u \cdot u - 2u \cdot u' + u' \cdot u' \\
&= 2 - 2\cos(\angle(u', u)) \\
&\le 2 - 2(1 - \tfrac{\angle(u',u)^2}{2}) \\
&= \angle(u', u)^2 \le \frac{1}{k^2}.
\end{aligned}
$$

Hence, the density of $G(\frac{1}{k}, t, u')$ on $x$ is lower bounded by,

$$
\left(\frac{1}{(2\pi)^{d/2}} k \exp(-5)\right)^{d-1} = cu_k.
$$

For every $t \in T$, we found a set of $u' \in N$ such that the density $G(\frac{1}{k}, t, u')$ assigns to $x$ is greater than $cu_k$. Since for some constant $c_d > 0$ depending only on $d$,

$$
\begin{aligned}
\mathbb{P}_{u' \sim \mathcal{U}(N)} \left\{u' \in C(u, \tfrac{1}{k})\right\} &= c_d \int_0^{1/k} \sin^{d-2}(x) dx \\
&\ge c_d \int_0^{1/k} \left(\frac{2}{\pi} x\right)^{d-2} dx \\
&= c_d \left(\frac{2}{\pi}\right)^{d-2} \frac{1}{d-1} \cdot \frac{1}{k^{d-1}},
\end{aligned}
$$

and since $x \in B$ was arbitrary, we indeed have,

$$
s_k = \inf_{x \in B} \mathbb{P}_{Q \sim \mathcal{U}(\mathcal{Q}_k)} \left\{Q^{d-1}(x) \ge cu_k\right\} = \Omega\left(\frac{1}{k^{d-1}}\right).
$$

This completes the proof of the claim. $\qquad\square$

With the three claims, applying Lemma G.1 allows us to conclude that the class of all Gaussians in $\mathbb{R}^d$ is not list learnable with $m(\alpha, \beta) = d - 1$ samples. This implies that the class is also not public-privately learnable with $m(\alpha, \beta, \varepsilon) = d - 1$ public samples. $\qquad\square$

## H  Statements and proofs for Section 7 – Learning when the Yatracos class has finite VC dimension

When the VC dimension of the Yatracos class of $\mathcal{Q}$ is finite, the following gives an upper bound on the number of samples required to non-privately learn $\mathcal{Q}$.

**Fact H.1** ([Yat85], [DL01, Theorem 6.4]). *Let $\mathcal{Q} \subseteq \Delta(\mathcal{X})$. Let $\mathcal{H}$ be the Yatracos class of $\mathcal{Q}$, and let $d = \mathrm{VC}(\mathcal{H})$. $\mathcal{Q}$ is learnable with*

$$m = O\left(\frac{d + \log(\frac{1}{\beta})}{\alpha^2}\right)$$

*samples.*

For some classes of distributions, the above bound is tight. For example, it recovers the $\Theta(\frac{d^2}{\alpha^2})$ sample complexity for learning Gaussians in $\mathbb{R}^d$ [AM18].

To prove Theorem 7.2 we employ the following result on generating distribution-dependent covers for binary hypothesis classes with public data.

**Fact H.2** (Public data cover [ABM19, Lemma 3.3 restated]). *Let $\mathcal{H} \subseteq 2^{\mathcal{X}}$ and $\mathrm{VC}(\mathcal{H}) = d$. There exists $\mathcal{A} \colon \mathcal{X}^* \to \{H \subseteq 2^{\mathcal{X}} : |H| < \infty\}$, such that for any $\alpha, \beta \in (0, 1]$ there exists*

$$m = O\left(\frac{d \log(\frac{1}{\alpha}) + \log(\frac{1}{\beta})}{\alpha}\right)$$

*such that for any $p \in \Delta(\mathcal{X})$, if we draw $\boldsymbol{X} = (X_1, \dots, X_m)$ i.i.d. from $p$, with probability $\geq 1 - \beta$, $\mathcal{A}(\boldsymbol{X})$ outputs $\widehat{\mathcal{H}} \subseteq 2^{\mathcal{X}}$ and a mapping $f \colon \mathcal{H} \to \widehat{\mathcal{H}}$ with*

$$p(h \triangle f(h)) \leq \alpha \qquad \text{for all } h \in \mathcal{H}$$

*(where for $A, B \subseteq \mathcal{X}$, $A \triangle B$ denotes the symmetric set difference $(A \setminus B) \cup (B \setminus A)$). Furthermore, we have $|\widehat{\mathcal{H}}| \leq \left(\frac{em}{d}\right)^{2d}$.*

We also use the following pure DP algorithm for answering counting queries on finite domains.

**Fact H.3** (SmallDB [BLR13], [DR14, Theorem 4.5]). *Let $\mathcal{X}$ be a finite domain. Let $\mathcal{H} \subseteq 2^{\mathcal{X}}$. Let $\alpha, \beta \in (0, 1]$ and $\varepsilon > 0$, There is an $\varepsilon$-DP randomized algorithm, that on any dataset $\boldsymbol{x} = (x_1, \dots, x_n)$ with*

$$n = \Omega\left(\frac{\log(|\mathcal{X}|)\log(|\mathcal{H}|) + \log(\frac{1}{\beta})}{\varepsilon\alpha^3}\right)$$

*outputs estimates $\hat{g} \colon \mathcal{H} \to \mathbb{R}$ such that with probability $\geq 1 - \beta$,*

$$\left|\hat{g}(h) - \frac{1}{n}\sum_{i=1}^n \mathbb{1}_h(x_i)\right| \leq \alpha \qquad \text{for all } h \in \mathcal{H}.$$

*Proof of Theorem 7.2.* We use our $m$ public samples from the unknown $p \in \mathcal{Q}$ to generate a public data cover $\widehat{\mathcal{H}}$ and mapping $f \colon \mathcal{H} \to \widehat{\mathcal{H}}$ courtesy of Fact H.2, selecting $m$ to target error $\frac{\alpha}{6}$ and failure probability $\frac{\beta}{3}$. Note that this implies that with probability $\geq 1 - \frac{\beta}{3}$, for every $h \in \mathcal{H}$, $|p(h) - p(f(h))| \leq p(h \triangle f(h)) \leq \frac{\alpha}{6}$.

Next, we consider the representative domain of $\mathcal{X}$ with respect to $\widehat{\mathcal{H}}$, denoted by $\mathcal{X}_{\widehat{\mathcal{H}}}$. In other words, for every unique behaviour $(\mathbb{1}_{\hat{h}}(x))_{\hat{h} \in \widehat{\mathcal{H}}} \in \{0, 1\}^{|\widehat{\mathcal{H}}|}$ induced by a point $x \in \mathcal{X}$ on $\widehat{\mathcal{H}}$, we include exactly one representative $[x]$ in $\mathcal{X}_{\widehat{\mathcal{H}}}$. By Sauer's lemma we can conclude that

$$|\mathcal{X}_{\widehat{\mathcal{H}}}| \leq \left(\frac{e|\widehat{\mathcal{H}}|}{d^*}\right)^{d^*}.$$

Then, we take our $n$ private samples $\boldsymbol{X} = (X_1, \dots, X_n)$ and map each point $X_i$ to its representative $[X_i] \in \mathcal{X}_{\widehat{\mathcal{H}}}$, yielding a dataset of $n$ examples $[\boldsymbol{X}]$ on the finite domain $\mathcal{X}_{\widehat{\mathcal{H}}}$. Note that for any $\hat{h} \in \widehat{\mathcal{H}}$, $\frac{1}{n}\sum_{i=1}^n \mathbb{1}_{\hat{h}}(X_i) = \frac{1}{n}\sum_{i=1}^n \mathbb{1}_{\hat{h}}([X_i])$. Hence when we run SmallDB (Fact H.3) on the input $[\boldsymbol{X}]$ over the finite domain $\mathcal{X}_{\widehat{\mathcal{H}}}$ with finite class $\widehat{\mathcal{H}}$, choosing $n$ large enough, we obtain $\hat{g} \colon \widehat{\mathcal{H}} \to \mathbb{R}$ such that with probability $\geq 1 - \frac{\beta}{3}$, $|\hat{g}(\hat{h}) - \frac{1}{n}\sum_{i=1}^n \mathbb{1}_{\hat{h}}(X_i)| \leq \frac{\alpha}{6}$ for all $\hat{h} \in \widehat{\mathcal{H}}$.

We also ensure $n$ is large enough so that we get the uniform convergence property on $\widehat{\mathcal{H}}$, which has VC dimension $d$, with the private samples. That is, for all $\hat{h} \in \widehat{\mathcal{H}}$, with probability $\geq 1 - \frac{\beta}{3}$, $|p(\hat{h}) - \frac{1}{n}\sum_{i=1}^{n} \mathbb{1}_{\hat{h}}(X_i)| \leq \frac{\alpha}{6}$.

As a post-processing of $\hat{g}$, our learner outputs

$$\hat{q} := \arg\min_{q \in \mathcal{Q}} \sup_{h \in \mathcal{H}} |q(h) - \hat{g}(f(h))|.$$

By the union bound, with probability $\geq 1 - \beta$, all of our good events occur. In this case, we have for all $h \in \mathcal{H}$,

$$|p(h) - p(f(h))| \leq \frac{\alpha}{6}$$

$$\left| p(f(h)) - \frac{1}{n}\sum_{i=1}^{n} \mathbb{1}_{f(h)}(X_i) \right| \leq \frac{\alpha}{6}$$

$$\left| \frac{1}{n}\sum_{i=1}^{n} \mathbb{1}_{f(h)}(X_i) - \hat{g}(f(h)) \right| \leq \frac{\alpha}{6}$$

which implies $|p(h) - \hat{g}(f(h))| \leq \frac{\alpha}{2}$. So for any $q \in \mathcal{Q}$,

$$|q(h) - p(h)| - \frac{\alpha}{2} \leq |q(h) - \hat{g}(f(h))| \leq |q(h) - p(h)| + \frac{\alpha}{2}$$

$$\implies \mathrm{TV}(q,p) - \frac{\alpha}{2} \leq \sup_{h \in \mathcal{H}} |q(h) - \hat{g}(f(h))| \leq \mathrm{TV}(q,p) + \frac{\alpha}{2}.$$

We have that

$$\sup_{h \in \mathcal{H}} |\hat{q}(h) - \hat{g}(f(h)| \leq \sup_{h \in \mathcal{H}} |p(h) - \hat{g}(f(h)| \leq \mathrm{TV}(p,p) + \frac{\alpha}{2} \leq \frac{\alpha}{2}.$$

Therefore,

$$\mathrm{TV}(\hat{q},p) \leq \sup_{h \in \mathcal{H}} |\hat{q}(h) - \hat{g}(f(h)| + \frac{\alpha}{2} \leq \alpha.$$

It can be verified that the choices of $m$ and $n$ in the statement of Theorem 7.2 suffice. $\qquad\square$

