# OpenReview forum: "Private Distribution Learning with Public Data: The View from Sample Compression"
_NeurIPS.cc/2023/Conference — NeurIPS 2023 spotlight_

### Official Review · Reviewer_7w9P · 2023-06-24

**Soundness:** 3 good
**Presentation:** 4 excellent
**Contribution:** 3 good
**Rating:** 8
**Confidence:** 3

**Summary:**

This paper studies differentially private learnability of probability distributions.
The results put forward are very interesting!
For example, they show that the public-private learnability of a class of distributions is connected to the existence of sample compression schemes as well as to an intermediate notion of learning they define.

**Strengths:**

Originality:
This work is original, to the best of my knowledge.

Quality:
Quality of results and topic choice are very good; see questions.

Clarity:
Writing is clear; see questions.

Significance:
The results are very significant for the learning community.

**Weaknesses:**

Not significant weaknesses detected.

**Questions:**

Are there any complexity-theoretic implications of your work?
Learning is connected to complexity lower bounds.
Is this the case for private learning?
Maybe some stronger connection exists?

Line 43:
Nice question!
Can you please discuss similar questions?

Theorem 1.1:
You should emphasize this result more, as it is a main important result.

Line 118:
Please define TV.
(This is in the appendix...)

Line 142:
Please define $SC$ (as "sample complexity?").

General comment:
Since you are not including proofs in the main body, maybe simplify the bounds you are presenting?
This will save space and thus allow for more explanations.

Section 5:
Can you please better explain the notion of distribution shifted?

Section 6 and Section 7:
Can you please add some more intuition here?

Maybe put limitations in the main body?

There is no conclusion section!
Why?
You could put open problems there.

This paper has so many novel results (proofs in appendix).
Impressive!

**Limitations:**

Yes.

---

> ### Author Rebuttal · Authors · 2023-08-09
>
> Thank you for thoughtful comments on our work. We are happy to hear the reviewer finds our results and topic of study interesting. We also thank the reviewer for their nice suggestions and questions.
>
> In what follows, we respond to specific points raised by the reviewer.
>
> —---------------------
>
> > Are there any complexity-theoretic implications of your work? Learning is connected to complexity lower bounds. Is this the case for private learning? Maybe some stronger connection exists?
>
> We are not aware of any. But we are generally unfamiliar with this line of work – could you elaborate further on what kind of connections you have in mind (and link some papers)? The only thing that comes to mind is [Dan16] – but since our learners are (a) not efficient; and (b) are for distribution learning, rather than PAC learning), it seems like this work is not related.
>
> > Line 43: Nice question! Can you please discuss similar questions?
>
> Sure. Our work examines the minimal amount of public data needed to render a class pure privately learnable. We can also examine this question in the approximate DP case. Note that here, it is open whether there is even a separation between learnable and approximate DP learnable.
>
> The VC bound of Theorem 1.5, although more general, is loose and is off “qualitatively” also: when applied to gaussians, it yields a $d^2/\alpha$ public sample complexity, though we know $d$ is possible, with no dependence on $\alpha$. It would be interesting to understand what qualities of a distribution admit $\alpha$-independent public sample complexity, beyond a not-too-illuminating “it has an $\alpha$-independent compression complexity”. The situation is quite different in binary classification: [ABM19] shows that any class that is not learnable under pure privacy learnable requires $1/\alpha$ public samples.
>
> Another interesting question is whether a small amount of public data can lead to algorithmic improvements (runtime efficiency, simpler algorithms), as opposed to sample complexity improvements. For Gaussian mean estimation, one public sample removes the need for coarse estimation in [HKM22] (their coarse estimation algorithm formulates the task as an SDP via sum-of-squares paradigm, and runs in polytime for a large polynomial). It would be interesting to see if with more, but o(non-private sample complexity) public data, we can replace other steps to devise simpler, faster algorithms.
>
> > Missing or unclear definitions.
>
> We’ll address these in the revised version.
>
> > Section 6 and Section 7: Can you please add some more intuition here?
>
> > Maybe put limitations in the main body?
>
> > There is no conclusion section! Why? You could put open problems there.
>
> Yes, in our revised manuscript, we’ll focus on better discussion of our results.
>
> In particular, we plan to add proof sketches of the reductions, explanations of the results in Section 6 and 7, as well as more comparison to previous results (particularly for Gaussian mixtures) and implications.
>
> We also plan to put limitations and a conclusion in the main body. We’ll use the extra space granted and also try to find some space by simplifying or deferring bounds to the appendix and removing repeated theorem statements.
>
> —---------------------
>
> If the reviewer has any additional comments or questions that have not been adequately addressed, we would be happy to speak more during the discussion phase.
>
> **References**
>
> [Dan16] Amit Daniely. Complexity theoretic limitations on learning halfspaces. STOC’16.
>
> [ABM19] Noga Alon, Raef Bassily, and Shay Moran. Limits of private learning with access to public data. NeurIPS’19.
>
> [HKM22] Samuel B. Hopkins, Gautam Kamath, and Mahbod Majid. Efficient mean estimation with pure differential privacy via a sum-of-squares exponential mechanism. STOC’22.

---

> > ### Comment · Area_Chair_gBuh · 2023-08-18
> >
> > Dear authors,
> >
> > Your message has been noted.
> > The decision on your paper will be based on my discussion with the reviewers.
> > We will reach out to your should we require further clarifications.
> >
> > Regards,

---

### Official Review · Reviewer_cweb · 2023-07-06

**Soundness:** 4 excellent
**Presentation:** 4 excellent
**Contribution:** 4 excellent
**Rating:** 7
**Confidence:** 3

**Summary:**

The authors consider the problem of learning a distribution in a known class using private samples under pure DP, as well as a number of public samples drawn from the same distribution. Some distribution classes, such as Gaussians, are not learnable using a finite number of private samples, or even with a finite number of private samples and a small number of public samples, but are learnable with private samples and a medium number of public samples.

The authors make progress towards understanding what classes of distributions are learnable with what number of public samples. Their main contribution is to establish that the following three are equivalent (up to constants): (1) A class of distributions is public-private learnable, i.e. learnable with m public samples and some finite number of private samples (2) A class of distributions is realizably compressible with m samples, i.e. given a finite number of samples from a distribution, one can encode the distribution into m samples and some finite number of auxiliary bits such that some decoder can learn the distribution using only the encoded information (3) A class is list learnable with m samples, i.e. there is an algorithm that takes m samples and outputs a finite list of distributions such that one distribution in this list is close to the target distribution. The authors also give concrete/non-trivial bounds on the finite number of samples/finite list size in the equivalence statement.

The above equivalence has a number of applications. For example, since mixtures of Gaussians are known to be compressible, the above equivalence immediately shows they are public-private learnable. More generally, since the class of mixtures (resp. products) of distributions from compressible classes is also compressible, mixtures (resp. products) of distributions from public-private learnable classes are also public-private learnable. While BKS22 established public-private learnability of Gaussians, the authors' result has several qualitative advantages, such as a stronger notion of DP and allowing arbitrarily small mixture probabilities. Similarly, the authors show that d-dimensional Gaussians are not learnable with d-1 public samples and any number of private samples, by showing that this implies list-learnability of d-dimensional Gaussians with d-1 public samples, which violates a lower bound.

The authors also extend the reduction from compressible classes to public-private learnable classes, to a reduction from robust compressible classes to classes that are public-private learnable, with out-of-distribution public samples and a private distribution that is only close to a member of the class.

Finally, the authors show that classes of distributions whose Yatracos class has VC dimension d and dual VC dimension d* are learnable with d private samples and d^2 times d* private samples.

**Strengths:**

Overall I feel the paper is quite strong. The main contribution of the paper, the equivalence of three properties of a class of distribution, is elegant and likely to have high impact, as distribution learning is obviously a very central and fundamental problem in learning theory, and using public data for private learning is becoming increasing popular. As an example of the potential for impact, just from the equivalence of these three classes, the authors are able to show a lower bound on the number of public samples needed to learn a Gaussian that is optimal up to an additive difference of 1, which was an open problem beforehand. I am not very confident in my knowledge of the field of distribution learning, but I suspect the elegance of the result could easily lend itself to understanding the public-private learnability of other important classes of distributions. In addition to this main contribution, there are a number of other contributions that are independently interesting. The paper is also technically interesting and novel. e.g, this paper is the first to use list-learning (which is related to, but not the same as list-decodable learning) in establishing upper and lower bounds for other definitions of learnability. The paper was enjoyable and easy to read - while there are several involved definitions needed to fully understand the results, the authors do a good job distilling the results down into a form that's easier to parse, without sacrificing too much detail.

**Weaknesses:**

I felt the main "weakness" of the paper is that there are many results the authors provide, and so the authors had to save most of the discussion of proof techniques for the appendix. In particular, I felt Sections 6 and 7 did not offer much more insight than the related sections of the introduction, and could be moved to the appendix in favor of e.g. a proof sketch for Prop 3.2, which would help the reader better understand the equivalence between the different definitions of learnability. Of course, such a weakness is likely to not exist in a camera-ready or arxiv version.

**Questions:**

-You mention that you do not optimize the private sample complexity - do you have a sense for how large the slack in the private sample complexities might be? (i.e., are they optimal up to log factors, small polynomials, large polynomials?)

**Limitations:**

Yes; in Appendix A the authors nicely state the limitations of their work.
N/A on negative societal impact, since it is a theory work on improving privacy-preserving algorithms.

---

> ### Author Rebuttal · Authors · 2023-08-09
>
> Thanks for the thoughtful comments on our work. We are glad the reviewer finds the problem we study to be relevant, and our results to have potential for impact.
>
> In what follows, we address specific points raised by the reviewer.
>
> —---------------------
>
>
> > Inadequate discussion of proof techniques in the main body.
>
> Noted. We’ll add proof sketches for the reductions (Proposition 3.2, 3.5, 3.6) and spend more time discussing results, in particular for Sections 6 and 7, since right now they are mostly repetitions of the theorem statements in the intro.
>
> > You mention that you do not optimize the private sample complexity. Do you have a sense for how large the slack in the private sample complexities might be? (i.e., are they optimal up to log factors, small polynomials, large polynomials?)
>
> What we were referring to there were the reductions to and from sample compression. They preserve public data complexity, but lose polynomial factors in the private sample complexity.
>
> For example: by applying the public-private learning/sample compression equivalence to obtain the generic “public-private learning implies public-private learning for mixtures” statement, if we apply it to a Gaussian public-private learning using $m = \tilde O(d)$ public and $n = \tilde O(d^2/\alpha^2 + d^2/\alpha\varepsilon)$ private samples, we get a $m = \tilde O(kd/\alpha)$ public, $n = \tilde O(d^2/\alpha^3 + d^2/\alpha^2\varepsilon)$ private public-private learner for Gaussian mixtures. This is a factor of $1/\alpha$ off from the sample complexity obtained from using the compression scheme for mixtures of Gaussians (Theorem 1.2).
>
> On another note, for Theorem 1.2 and Corollary 4.1, in the specific regime where $\varepsilon>\alpha$, the private sample complexities are tight up to log factors, based on non-private lower bounds of $\Omega(d^2/\alpha^2)$ for a single gaussian and $\tilde \Omega(kd^2/\alpha^2)$ for mixtures. This would be the case even with more than $O(d)$, but $o($non-private sample complexity$)$ public samples.
>
> —---------------------
>
> If the reviewer has any additional comments or questions that have not been adequately addressed, we would be happy to speak more during the discussion phase.

---

> > ### Comment · Reviewer_cweb · 2023-08-11
> >
> > Thank you for the detailed response! After seeing the other reviews and rebuttals as well, I am planning on keeping my rating of accept.

---

### Official Review · Reviewer_rJcv · 2023-07-07

**Soundness:** 4 excellent
**Presentation:** 3 good
**Contribution:** 3 good
**Rating:** 6
**Confidence:** 4

**Summary:**

Continuing a line of work recently initiated by Bie, Kamath, and Singhal (NeurIPS '22), this submission studies differentially private distribution learning (a.k.a. hypothesis selection) in the presence of public data. The main result is a collection of quantitative equivalences between this task and the problems of sample compression and list learning. More specifically, the paper shows that for any class of distributions Q, the following are (roughly) equivalent:
1) Q admits a sample compression scheme using m samples and compression size t
2) Q is pure $\varepsilon$-DP learnable using m public samples and t private samples
3) Q is list learnable using m samples and list size exp(t).

As applications of this set of equivalences, the paper proves new upper and lower bounds on the sample complexity of pure public-private learning. These include new algorithms for learning Gaussians, mixture distributions, and product distributions via sample compression, as well as a lower bound on the public sample complexity of pure public-privately learning a single Gaussian that follows from a lower bound for list learning. The paper also works out a connection between robust sample compression and agnostic/distribution-shifted public-private learning, and a pure public-private learner for classes whose Scheffe sets have finite VC dimension.

**Strengths:**

- The equivalences shown in this paper provide a nice set of perspectives on the public-private distribution learning problem, both conceptually and "practically" in terms of offering new avenues for proving upper and lower bounds.

- The proofs of these equivalences are simple, conceptually clear, and illuminating.

- I'd say the main technical contribution in this paper is a lower bound showing that it is impossible to list-learn d-dimensional Gaussians using fewer than d samples. (This in turn implies a lower bound of public-private learning.) This is a sharp result that matches an upper bound from BKS'22. The proof goes through a general lower bound technique for list-learning, called a "no-free-lunch" theorem in this paper, roughly based on exhibiting collections of distributions that are not close in TV distance, but easily confused using a small number of samples.


**Weaknesses:**

- The characterizations shown in this work all apply to pure differential privacy. This is good for upper bounds, but is a major limitation of the most technically-involved contribution of the paper on lower bounding list- / public-private learnability of Gaussians. In particular, the conversion from a public-private learner to a list-learner depends crucially on pure differential privacy; while it can probably be extended to a quantitatively weaker result for concentrated DP, it doesn't say anything about, say, Renyi or approximate DP.

- The paper's conversion from a public-private learner to a list-learner is not quite constructive, as it relies on a covering-based characterization of pure private distribution learning. However, this characterization can probably be unrolled into an algorithmic construction of a cover using the usual technique of running the DP algorithm repeatedly on a fixed dataset using many sequences of coin tosses. Also, none of the equivalences proved preserve computational efficiency in general.

**Questions:**

- Lines 235-239 offer a comparison to the previous result of BKS'22 on learning mixtures of Gaussians, but isn't easy to make sense of without a brief (informal) statement of what their result actually is.

- Line 500: Should be $\tau_0 = \tau(\alpha/6, \beta/2)$

- While ultimately short, I found the contradiction-based proof of Proposition 3.5 unnecessarily hard to follow for this reason. Why not just do a direct proof along the following lines, using roughly the same calculations? For every distribution $p \in \mathcal{Q}$, accuracy of $\mathcal{A}$ implies that whp over $\widetilde{X} \sim p^m$, we have $p \in Q_{\widetilde{X}}$. This latter event implies $p$ is close to $\mathcal{L}(\widetilde{X})$, which is the success condition of the list learner.

- I didn't understand why Lemma F.1 is called a "no free lunch" theorem. To my mind, a no free lunch result rules out the existence of learning algorithms that are simultaneously optimal for a sufficiently broad class of problems. Did the authors have in mind a different context in which this name makes sense to describe this result?

**Limitations:**

Some limitations (overlapping with weaknesses described above) are discussed in the appendix.

---

> ### Author Rebuttal · Authors · 2023-08-09
>
> Thank you for the thoughtful comments and suggestions. We are glad the reviewer finds that our work offers some nice perspectives on the public-private learning problem. Indeed, we acknowledge the limitations pointed out by the reviewer: (a) that our equivalence only holds for the pure DP setting and (b) we do not address issues pertaining to computational efficiency.
>
> Both aspects are important directions for future work; indeed for one of the tasks studied – density estimation of arbitrary Gaussian mixtures – the authors are not aware of any poly-time algorithm for the task, even in the non-private setting.
>
> In what follows, we reply to specific points raised by the reviewer.
>
> —---------------------
>
> > This characterization can probably be unrolled into an algorithmic construction of a cover using the usual technique of running the DP algorithm repeatedly on a fixed dataset using many sequences of coin tosses.
>
> That’s really interesting. We would be happy to learn more about this – would it be possible to elaborate further/link an example of a paper that uses this technique?
>
> > Lines 235-239 offer a comparison to the previous result of BKS'22 on learning mixtures of Gaussians, but isn't easy to make sense of without a brief (informal) statement of what their result actually is.
>
> Noted. We’ll add a more comprehensive discussion of this result and other related results.
>
> > Line 500: Should be 𝜏 = 𝜏(ɑ/6, β/2)
>
> Thanks, we fixed this.
>
> > I found the contradiction-based proof of Proposition 3.5 unnecessarily hard to follow for this reason. Why not just do a direct proof
>
> Good point, we’ll change it to a direct proof.
>
> > I didn't understand why Lemma F.1 is called a "no free lunch" theorem.
>
> The context we had in mind is from the VC lower bound results for PAC learning, i.e., Theorem 5.1 from [SSBD14]. In that result, using the flexibility of a hypothesis class, a class of hard instances are constructed such that any algorithm that doesn’t see enough samples must fail on one of them. Hence, there is no universal learner (free lunch).
>
> Similarly, in our Lemma F.1, we use the flexibility of the class $\mathcal Q$ to construct a sequence of classes of hard instances ($\mathcal Q_k$). Each $\mathcal Q_k$ rules out the existence of a list learning algorithm that takes few samples, outputs a list of size $\leq k$, and succeeds on every $p$ in $\mathcal Q_k$. This rules out the existence of any finite list learner that sees few samples and succeeds on every $p$ in $\mathcal Q$.
>
> We hope this clarifies things. We’ll include this discussion in the revised version of the manuscript.
>
> —---------------------
>
> If the reviewer has any additional comments or questions that have not been adequately addressed, we would be happy to speak more during the discussion phase.
>
> **References**
>
> [SSBD14] Shai Shalev-Shwartz and Shai Ben-David. Understanding machine learning: From theory to algorithms. Cambridge University Press, 2014.

---

> > ### Comment · Reviewer_rJcv · 2023-08-14
> >
> > Thank you for your response, which indeed helps clarify and confirms my recommendation to accept.
> >
> > >> This characterization can probably be unrolled into an algorithmic construction of a cover using the usual technique of running the DP algorithm repeatedly on a fixed dataset using many sequences of coin tosses.
> >
> > > That’s really interesting. We would be happy to learn more about this – would it be possible to elaborate further/link an example of a paper that uses this technique?
> >
> > Of course. After thinking about it a bit more carefully, I think what you actually get is a randomized analog of a cover, which in turn induces a randomized list learner. (So depending on your personal view of what counts as constructive, this may or may not improve on what you have...but at least it may be a start for further investigation.)  Starting from line 514 of your supplementary material (and simplifying notation a bit because the OpenReview editor is bugging out on me), let $\mathcal{Q}$ be learnable by some $\varepsilon$-DP algorithm $A$ using $n$ samples. Let $p \in \mathcal{Q}$ be an arbitrary member of the class, and define $G_p$ to be the set of all distributions that are $\alpha$-close to $p$ in TV distance. By accuracy of the DP algorithm, $\Pr_{X \sim p^n, A}[A(X) \in G_p] \ge 9/10$, so by averaging, there exists a sample $S$ of size $n$ such that $\Pr_A[A(S) \in G_p] \ge 9/10$. By group privacy, this implies $\Pr_A[A(0^n) \in G_p] \ge 9/10 \cdot e^{-\varepsilon n}$. Thus, by running $A(0^n)$ some number $\ell = O(e^{\varepsilon n})$ times using independent coin tosses, with constant probability, one obtains a list $L$ of distributions containing at least one distribution $\widehat{p}$ that is $\alpha$-close to $p$. (The induced distribution over such lists is what I would call a "randomized cover", and the algorithm that outputs a list from this distribution is a randomized list-learner.)
> >
> > This idea goes at least as far back as https://arxiv.org/abs/1402.2224, Lemma 3.15, where it was used to show that any pure-DP PAC learner for a concept class $\mathcal{C}$ induces a "probabilistic representation" for $\mathcal{C}$. Indeed, in the PAC setting, one can think of a probabilistic representation as a randomized list learner that ignores the sample. It's interesting to note that in the PAC setting, there's a gap between what one can achieve with a deterministic representation (the analog of a standard cover) vs. a probabilistic representation, whereas in the distribution learning world, your results imply that there's no such gap.
> >
> > > The context we had in mind is from the VC lower bound results for PAC learning, i.e., Theorem 5.1 from [SSBD14]. In that result, using the flexibility of a hypothesis class, a class of hard instances are constructed such that any algorithm that doesn’t see enough samples must fail on one of them. Hence, there is no universal learner (free lunch).
> >
> > Thanks for the explanation. My understanding of the reason why the result you mention is called "no free lunch" is indeed because it rules out the existence of a universal learner for the class of _all_ concepts over $\mathcal{X}$. This motivates the need to make prior knowledge assumptions, e.g., assuming a smaller concept class than the space of all concepts, as elaborated in their Section 5.1.1. Said another way, the supposed "free lunch" in question would be a universal learner that does not need to make any assumptions about the structure of the target concept, and is ruled out by a sample complexity lower bound proportional to the domain size.
> >
> > Meanwhile, in your case, the lower bound your prove is _already_ for a pre-specified class of distributions $\mathcal{Q}$. So the sense of a "universal learner" that you rule out is already one that is only universal with respect to $\mathcal{Q}$ -- which is a much weaker sense of universality than the one in the SB NFL theorem. Given the context of the SB NFL, my guess for a public-private distribution-learning NFL theorem would say something like, "Every public-private learner for the class of all distributions over a finite domain $\mathcal{X}$ requires at least ... samples."
> >
> > Of course, as you point out, there are technical similarities in the proofs. But I'd argue that a better analogy for your result is the implied VC-dimension lower bound for PAC learning; as you alluded to, one can of course derive this from the SB NFL by embedding into a high-capacity class, but the VC-dimension statement is one about class-specific learning. If anything, I think calling your result an NFL theorem undersells that it's in fact able to say something on a class-by-class basis.

---

> > > ### Author Response · Authors · 2023-08-21
> > >
> > > Thanks for the detailed reply!
> > >
> > > Very nice! Thanks for the reference and the explanation. Yes, I believe this will go through and give us constructive learners for Theorem 4.5 and Theorem 4.6. We can add a note pointing to and summarizing this discussion. It also motivates considering randomized compression schemes, which might be interesting to play around with.
> > >
> > > The point on NFL is well-taken; similarities between the lemma and Theorem 5.1 of [SSBD14] are on technical side rather than conceptual, and the NFL naming choice is a conceptual one. Indeed Gaussianity is a strong inductive bias yet we still have a negative result, so its more "no-reasonably-priced-lunch". To prevent confusion we'll refer to the lemma as a lower bound employing a NFL-style argument.

---

### Official Review · Reviewer_PbeY · 2023-07-09

**Soundness:** 4 excellent
**Presentation:** 3 good
**Contribution:** 3 good
**Rating:** 7
**Confidence:** 3

**Summary:**

In public-private learning, an algorithm gets two sets of i.i.d. samples from an unknown distribution $q$ from a class $\cal{Q}$ (say, the first set of size $m$ and the second one of size $n$), but is required to preserve DP only with respect to the second set.
This research direction is about understanding what are the (minimal) regimes of $m$ and $n$ that allows to learn a distribution from $\cal{Q}$. The interesting regimes are when $m$ is much smaller than the sample complexity required to learn without privacy, and $n$ is smaller than what is required for learning under (full) DP.
This research direction has been initiate by Bie, Kamath and Singhal (NeurIPS 22’, BKS22), who showed that a relatively small amount of public samples suffices to improve the sample complexity of privately learning a single Gaussian and a mixture of Gaussians.
This work makes a significant step towards characterizing public-private learning by proving that it is equivalent (in terms of sample complexity) to two other notions: (1) Compressible Learning (Ashtiani et al., J.ACM 2020, ABDH+20) and (2) List-Learning. In more detail (ignoring running times and low order terms and constants in the sample complexity parameters):

Proposition 3.2: $(\tau, t, m)$-compressible learning implies $(m,n = t + \tau \log m)$-public-private learning, where the former means that we encode $m$ samples using a $\tau$ size subset plus an auxiliary string of size $t$, and this information suffices for estimating the distribution.

Proposition 3.5: $(m,n)$-public-private learning implies $(m,\ell=\exp(n))$-list learning, where the latter means that using $m$ samples we can output a list of size $\ell$ of candidate distributions that at least one of them is a good estimation to the true distribution.

Proposition 3.6: $(m,\ell)$-list learning implies $(\tau=m, t = \log(\ell), m)$-compressible learning.

The main applications comes from the equivalence to compressible learning, since [ABDH+20] proved that learning a Gaussian or a mixture of Gaussians is compressible learnable with relatively small amount of samples, so using this connection they deduce an improved sample complexity upper bounds for these tasks under private-public learning, as well as a tight lower bound of the number of public samples that is required for learning a single Gaussian. The upper bound on public-private learning a mixture of Gausians is actually an upper bound for learning any mixture of distributions for a given class $\cal{Q}$ as a function of the sample complexity that is required to learn a single distribution from $\cal{Q}$, as [ABDH+20] proved that for compressible learning, learning a $k$ mixture requires (esseitially) $k$ times the sample complexity for a single distribution.

In section 5 they also handle the agnostic case where the samples comes from a distribution that is only close to a one in the class, a setting which was also considered in the previous work of [BKS22]. They prove (Theorem 5.2) that robust compression implies agnostic public-private learning, and deduce an upper bound on learning a Gaussian in this case.


**Strengths:**

The equivalences between public-private learning to the other notions of learning are very interesting, and the progress from the work of [BKS22] (both quantitative and qualitative) is significant. Despite the significant limitations (described below), in my opinion, this is a good submission which is above the acceptance threshold. Therefore, I recommend acceptance.

**Weaknesses:**

As mentioned in the limitations section (Section A, Supplementary material), the reductions between the different notions are inefficient, and sometimes not even constructive (e.g., in the reduction from list learning to public-private learning). So all the new upper bounds of this paper are inefficient, and it is still left open whether we can construct efficient public-private learning algorithms with the improved samples complexities achieved in this paper.
While the results of [BKS22] are more restricted and weaker in terms of sample complexity, they are computationally efficient (unlike this work), so if we are only interested in computationally efficient learners, this work does not make any progress except the lower bound which holds (in particular) for efficient learners.


**Questions:**

No questions.


One suggestion:

I think it is worth adding short proof sketches of the reductions (Propositions 3.2, 3.5, 3.6) in the main body. The proofs are not long, and I'm sure you can explain the idea in each proof using a few sentences.

Minor comments:

(1) Theorem 1.1, item 2: "$\cal{Q}$ is has".

(2) Theorem 1.3: robust compression is not defined at this point.

(3) Definition 2.5: In ABDH+20, $f_q$ is explicitly defined by outputting an $\tau_0$-subset of samples, and a short string. Here, you just claiming the existence of such $f_q$, which is strange to me since, for example, if $f_q$ is randomized and depends on $q$, it could simply ignore the input samples (which are sampled from a distribution that is only close to $q$) and just use fresh samples from $q$.

(4) Corollary 4.2: I think it is worth comparing the sample complexity to existing efficient and fully DP methods like https://arxiv.org/abs/2303.04288 and https://arxiv.org/pdf/2112.14445.pdf (BKS22 only compared their result to KSSU19 https://arxiv.org/abs/1909.03951 which is significantly inefficient compared to the first two results I mentioned).

**Limitations:**

The authors present the limitations in Section A (supplementary material).

---

> ### Author Rebuttal · Authors · 2023-08-09
>
> Thank you for the thoughtful comments and suggestions. We are glad that the reviewer finds the connections we present between public-private learning and other notions of learning interesting. Indeed, our investigation only answers questions regarding the statistical complexity of public-private learning. Finding efficient algorithms for these tasks is not addressed by our work, and is an important open question.
>
> In what follows, we address specific points raised by the reviewer.
>
> —---------------------
>
> > Adding short proof sketches for the reductions.
>
> Yes, we’ll add proof sketches for the reductions in the revised manuscript.
>
> > Typos and organization
>
> Thanks for pointing out typos and the out of order definition. Theorem 1.3 requires some setup, so we’ll introduce its consequence (agnostic and distribution shifted learner for Gaussians) and leave the statement for the main body.
>
> We’ll also move the limitations to the main body in the revised version, with the extra space granted.
>
> > Definition 2.5: In ABDH+20, $f_q$ is explicitly defined by outputting an 𝜏-subset of samples, and a short string. Here, you just claiming the existence of such $f_q$, which is strange to me since, for example, if $f_q$ is randomized and depends on $q$, it could simply ignore the input samples (which are sampled from a distribution that is only close to $q$) and just use fresh samples from $q$.
>
> Yes, we missed a condition in the definition. The $f_q$ defined should have the property that on all samples $S \in \mathcal X^m$, $ f_q(S)$ is a subset of $S$. We’ll fix this in the revised version.
>
> > Corollary 4.2: I think it is worth comparing the sample complexity to existing efficient and fully DP methods like https://arxiv.org/abs/2303.04288 and https://arxiv.org/pdf/2112.14445.pdf (BKS22 only compared their result to KSSU19 https://arxiv.org/abs/1909.03951 which is significantly inefficient compared to the first two results I mentioned).
>
> We’ll do a more comprehensive discussion of results in the updated version.
>
> All three of these are looking at parameter estimation with assumptions on the underlying Gaussian mixture, whereas we focus on density estimation of arbitrary Gaussian mixtures (and we are not aware of any $\text{poly}(k,d,\alpha,\beta)$ algorithms even in the non-private case).
>
> [AAL23] uses [MV10] as a black box, and inherits its unoptimized and very large polynomial sample complexity. We believe the exponent of the polynomial is something like $\approx 300$, but we are not exactly sure.
>
> —---------------------
>
> If the reviewer has any additional comments or questions that have not been adequately addressed, we would be happy to speak more during the discussion phase.
>
> **References**
>
> [AAL23] Jamil Arbas, Hassan Ashtiani, and Christopher Liaw. Polynomial time and private learning of unbounded gaussian mixture models. ICML’23.
>
> [MV10] Ankur Moitra and Gregory Valiant. Settling the polynomial learnability of mixtures of gaussians. FOCS’10.

---

> > ### Comment · Reviewer_PbeY · 2023-08-15
> >
> > Thank you. I read the rebuttal and have no questions.

---

### Decision · Program_Chairs · 2023-09-21

**Decision:**

Accept (spotlight)

**Comment:**

The reviewers unanimously agreed that this work provides insightful perspectives on the public-private distribution learning problem, both conceptually and algorithmically in terms of offering new avenues for proving upper and lower bounds.